

# Phosphorus solubility in aerosol particles related to particle sources and atmospheric acidification in Asian continental outflow

Jinhui Shi[1,2], Nan Wang[1], Huiwang Gao[1,2], Alex R. Baker[3], Xiaohong Yao[1,2], Daizhou Zhang[4]

[1]Key Laboratory of Marine Environmental Science and Ecology, Ocean University of China, Ministry of Education of China, Qingdao 266010, China
[2]Laboratory for Marine Ecology and Environmental Science, Qingdao National Laboratory for Marine Science and Technology, Qingdao 266237, China
[3]Centre for Ocean and Atmospheric Sciences, School of Environmental Sciences, University of East Anglia, Norwich, NR4 7TJ, UK
[4]Faculty of Environmental and Symbiotic Sciences, Prefectural University of Kumamoto, Kumamoto 862-8502, Japan

*Correspondence to*: Jinhui Shi (engroup@ouc.edu.cn)

**Abstract.** The continent-to-ocean supply of phosphorus (P) in the soluble state, recognized as bioavailable P, via the atmosphere is hypothesized to be crucial to the biological cycle in offshore surface seawater. To investigate the solubility of P in aerosol particles moving towards the Northwestern Pacific from the Asian continent, we measured the total P (TP), total dissolved P (TDP) and dissolved inorganic P (DIP) in aerosols at Qingdao (36°06′ N, 120°33′ E), a coastal city in eastern China. The samples were collected in December 2012 and January 2013 (winter) and in March and April 2013 (spring), when the middle latitude westerly wind was prevailing. On average, P solubility, i.e., the ratio of TDP to TP, was $32.9 \pm 16.7$ % in winter and $21.3 \pm 9.8$ % in spring, and the TP concentrations in the two seasons were similar. This seasonal solubility difference is attributed to the aerosol sources containing the P. Particles in winter were predominantly anthropogenic particles from local and regional areas, and particles in spring were significantly influenced by natural dust from the arid and semiarid areas in the inland part of the continent. Moreover, acidification processes associated with the formation of sulfate and nitrate in the winter samples enhanced P solubility, suggesting that the P in anthropogenic particles was more susceptible to the production of acidic species than that in natural dust particles. There was a strong positive correlation between P solubility and relative humidity (RH). P solubility was usually less than 30 % when RH was below 60 %, even when the content of acidic species and/or anthropogenic particles in the aerosols was high, suggesting humidity had a critical role in the production of TDP. In addition, the proportion of DIP in TDP was high when the particles were predominantly anthropogenic, and the proportion of dissolved organic P (DOP; quantified as TDP-DIP) in TDP was high when the particles were dominated by natural dust. These results indicate that, as the contents of bioavailable P in Asian continent outflows are closely dependent on the aerosol particle origins, atmospheric acidic processes could convert P into a bioavailable state under certain meteorological conditions. Therefore, the recent severe air pollution over East Asia might have enhanced the input of bioavailable P to downwind marine areas.



# 1 Introduction

Phosphorus (P) is one of the limiting nutrients for primary production in marine ecosystems, affecting marine phytoplankton growth, community structure composition and nitrogen fixation (Elser et al., 2007; Paytan and McLaughlin, 2007; Peñuelas et al., 2013). Atmospheric P deposition has been shown to induce the growth of phytoplankton in surface seawater outside estuary areas, especially in offshore areas and regions where P limits phytoplankton growth (Paytan and McLaughlin, 2007; Mackey et al., 2012a). Recent studies have found that the atmospheric input of anthropogenic N to the North Pacific and its marginal sea is increasing (Kim et al., 2013; Kim et al., 2014), while the P concentration in the surface seawater has been declining for nearly 40 years (Kodama et al., 2016), indicating that the nutrient structure of marine ecosystems is changing to P limitation.

Atmospheric P is derived from natural and anthropogenic sources, including mineral dust, sea salt, primary biogenic sources, volcanic eruptions, biomass burning, fossil fuel combustion, and agricultural fertilizers (Mahowald et al., 2008; Anderson et al., 2010; Tipping et al., 2014; Weinberger et al., 2016). It has been widely recognized that mineral dust is the main source of atmospheric P. Based on a combination of model simulations and field observations, Mahowald et al. (2008) reported that mineral aerosols contributed to approximately 82 % of atmospheric P at a global scale, while anthropogenic P contributed to approximately 5 % of atmospheric P globally. Recent studies have highlighted the contributions of anthropogenic sources. Weinberger et al. (2016) suggested that coal combustion is a substantial source of atmospheric P and that the relative content of P in coal fly ashes could be as high as 3500 μg g$^{-1}$. Srinivas and Sarin (2015) studied the sources of P in the atmospheric aerosols of Bengal Bay in the northern Indian Ocean and found that 75 % of aerosol P was from agricultural fertilizers and biomass burning. A recent estimation by Wang et al. (2015) showed that combustion-related emissions could contribute more than 50 % of the global atmospheric P. These results suggest that anthropogenic emissions of P have likely increased considerably in recent years and that the states of P in atmospheric aerosols need to be carefully quantified.

P solubility, i.e., the ratio of TDP to TP, is usually used to characterize the bioavailability of atmospheric P (Anderson et al., 2010) because the soluble fraction of P in aerosols is believed to be the form that can be directly assimilated by marine phytoplankton (Mackey et al., 2012b). Unfortunately, observational data on the solubility of atmospheric P are very limited, and the available data show a wide range of P solubility, ranging from 2 % to 100 % in different sea areas (e.g., Baker et al., 2006a, b; Mahowald et al., 2008).

Dissolved P in aerosols includes dissolved inorganic P (DIP) and dissolved organic P (DOP). Only DIP has been frequently analyzed by previous studies of P solubility in aerosols (Srinivas and Sarin, 2015) because DIP is considered directly available to marine phytoplankton and the predominant component of total dissolved P (Mahowald et al., 2008). However, some studies have found that DOP can be converted to bioavailable P by enzymatic reactions (Mackey et al., 2012a) and accounts for 20-83 % of TDP in global oceanic atmospheric depositions (Kanakidou et al., 2012), although the contribution of DOP to TDP is sometimes very low (Izquierdo et al., 2012).

P solubility in aerosols from different sources varies greatly and is generally low in particles from mineral sources and high in particles from anthropogenic sources. In the Mediterranean atmosphere, P solubility ranges from 2-20 % with an average of



10 % in aerosols affected by Saharan dust, whereas P solubility ranges from 30-79 % with an average of 50 % in aerosols with anthropogenic sources (Herut et al., 2002). Similar results were also observed in the Atlantic atmosphere, where the P solubility in aerosols affected by Saharan dust is approximately 8 % and can be up to 87 % in aerosols originating from terrestrial air masses in South America (Baker et al., 2006a; 2006b). In the South China Sea, P solubility in aerosols related to biomass

combustion ranges from 50 ± 14 % (Hsu et al., 2014). These results imply that an increase in the relative amounts of aerosols from anthropogenic sources might lead to the increase of bioavailable P. In addition, atmospheric acidic processes associated with anthropogenic pollutants may transform unreactive P to bioavailable P. Recent model studies predict that acid dissolution process increases the fraction of bioavailable P from ~10% globally at labile pools to 42% in the Pacific Ocean, with the mean value of 22% in global marine atmosphere (Herbert et al., 2018).

The Yellow Sea is a marginal sea of the Pacific Ocean that is frequently affected by both Asian dust from arid and semiarid areas in inland regions of the Asian continent and anthropogenic pollutants from urban agglomerations in northern China (e.g., Zhang and Gao, 2007; Zhang et al., 2018; Shi, et al., 2012; Wang, et al., 2013). For example, a dust storm that occurred in May 2017 moved across the North Pacific within a week and deposited 5.3 Tg of aeolian dust across the North Pacific Ocean (Zhang et al., 2018). In recent years, anthropogenic emissions have rapidly increased with the acceleration of industrialization

and urbanization over North China. Prevailing westerly winds in the Northern Hemisphere middle latitudes can carry anthropogenic aerosols travelling over the North Pacific (Lyu et al., 2017; Joos et al., 2017). Over the last 30 years, the structure of nutrients in the Yellow Sea water has exhibited an alternation from N limitation to potential P limitation (Wei et al., 2015). However, there are no data on P solubility in the atmospheric aerosols outflowing from the Asian continent towards the Yellow Sea and the northern Pacific, which hinders further understanding of the influence of P deposition on the biological cycles in

the surface seawater of those ocean areas.

In this study, we investigated TP and TDP, including DIP and DOP, in aerosols collected at Qingdao. The city of Qingdao is located in the coastal area of the Yellow Sea. In winter and spring, when westerly winds prevail, aerosol particles originating from the Asian continent, including natural dust and anthropogenic particles, are frequently blown from the continent and enter the marine air over the Yellow Sea and subsequently over the northern Pacific (Zhang et al., 2005; Joos et al., 2017; Qi et al.,

2018; Zhang et al., 2018). The objectives of this paper are to characterize (1) the concentrations of TP, TDP, DIP and DOP in aerosols in the coastal areas of the Yellow Sea, (2) the relative contribution of DOP and DIP to TDP, (3) P solubility in aerosols in the Asian continental outflow, and (4) the influences of dust loads, particle sources, atmospheric acidification and ambient relatively humidity on P solubility.

## 2 Methods

### 2.1 Sample collection

Aerosol samples of total suspended particulates (TSP) were collected between December 1, 2012 and January 31, 2013 (winter) and between March 1 and April 30, 2013 (spring). The aerosol sampler used for collection was a high-volume TSP sampler



(KC-1000, Qingdao Laoshan Elec. Inc., China) set up on the roof of a building at the Ocean University of China in Qingdao (36°06′ N, 120°33′ E). The roof was approximately 65 m above sea level, and the building was < 1000 m from the coast of the Yellow Sea. In total, 112 samples were collected, of which sixty were collected in the winter period and fifty-two were collected in the spring period. Each aerosol sample was collected onto an acid-washed fiber filter (Whatman-41, 20.3 cm ×

25.4 cm) at a flow rate of 1.05 m$^3$ min$^{-1}$ for 24 h, and the total sampling air was approximately 1500 m$^3$. Operational filter blanks were also collected. All samples were sealed in polyethylene bags and stored in a freezer at -20 ℃ until subsequent analyses.

During the sampling periods, the particle concentrations in six size ranges were measured using a laser optical particle counter (OPC, ARTI Model HHPC-6). The OPC had 6 channels (i.e., 0.3, 0.5, 0.7, 1.0, 2.0 and 5.0 μm) and was run at a flow

rate of 2.83 L·min$^{-1}$ with a 15 min time resolution. Meteorological conditions, including temperature, relative humidity, wind speed and direction, and visibility were obtained from the Micaps meteorology data provided by the China Meteorological Administration. The 24 h average values of these variables during the sampling periods are illustrated in Fig. S1.

## 2.2 Sample analysis

The sample filters and blank filters were subdivided into portions using an acid-cleaned ceramic knife under a class-100

laminar flow hood. For the analysis of TP, a piece of the filter was digested with 4 mL of 70 % HNO$_3$ and 1 mL of 49 % HF (all ultra-pure grades) in a high-pressure Teflon jar at 180 ℃ for 48 h. Then, the digested sample solution was heated on an electrothermal board at 160 ℃ until the acid fumes dissipated. After being cooled to room temperature, the residue was redissolved, transferred to a colorimetric tube, and made up to a volume of 25 mL for determination of TP. For the analysis of soluble P, another piece of the sample filter was ultrasonically extracted with 15 mL Milli-Q water (≥ 18.2 MΩ·cm) for 20

min at 0 ℃. The extract was then filtered through a microporous membrane (pore size of 0.45 μm). The extraction was repeated once, and then the filter was rinsed three times with Milli-Q water. All the extractive solutions were combined and finally made up to a volume of 50 mL. The filtered extract was divided into two equal parts: one directly for the determination of DIP and the other autoclaved with 2.5 mL alkaline potassium persulfate solution (0.375 mol L$^{-1}$ NaOH, 0.185 mol L$^{-1}$ K$_2$S$_2$O$_8$ and 0.484 mol L$^{-1}$ H$_3$BO$_3$ mixed solution) at 120 ℃ for 30 min for the determination of TDP. DOP was obtained by the difference

between TDP and DIP.

P in the prepared sample solutions was measured by the molybdenum blue technique with colorimetric detection. A color reagent comprising a mixture of sulfuric acid, ammonium molybdate, ascorbic acid and antimony potassium tartrate was added into the solutions, and P in the solutions was quantified using a UV spectrophotometer (T-6 new century, Beijing General analysis) with a 5-cm quartz cell at 880 nm once the color was developed. The detection limit was 0.07 μmol L$^{-1}$ (approximately

0.6 ng m$^{-3}$), defined as three times the standard deviation of the blanks. The values for the blanks (i.e., blank filters and reagents) were below the detection limit. The relative standard deviations of replicate analysis of the sample extracts were within 3 %. The accuracy of the TP analysis procedure used in this study was checked using a soil sample standard reference material as a substitute (GBW07408, provided by the Geophysical and Geochemical Survey Institute, China). GBW07408 was included in



all analytical runs with the same treatment as sample filters, and the P recoveries were 95-105 % with an average of 98 % (n = 10).

Total and soluble trace elements, including Al, Fe, Mn, Ba, Ca, Zn, Ni, As, Cd, Pb and K, in samples were analyzed using an Agilent 7500c octopole-based inductively coupled plasma mass spectrometer (ICP-MS). Water-soluble inorganic ions, including $Na^+$, $K^+$, $Mg^{2+}$, $NH_4^+$, $Ca^{2+}$, $Cl^-$, $NO_2^-$, $NO_3^-$ and $SO_4^{2-}$, were analyzed using a Dionex ICS-3000 ion chromatograph. The full details of the sample extraction and analytical procedures for aerosol trace element and ion analyses are described in Shi et al. (2012, 2013).

## 2.3 Aerosol-specific surface area calculation

The specific surface areas of aerosol particles were calculated to investigate the possible dependence of P solubility on chemical conversions occurring in the aerosol surface layer. The specific surface area of an aerosol sample was defined as the ratio of the surface area of the particles to the particle mass loading. The aerosol surface area was estimated from the size-segregated number concentrations of aerosol particles measured by the optical particle counter in the sample collection period. We made a simplistic assumption of spherical particles, and hence:

$$S_i = \sum_{j=1}^{n} \left[ C_{i,j} \cdot 4\pi \cdot \left( \frac{d_j}{2} \right)^2 \right] \tag{1}$$

where $S_i$ is the total surface area of the particles in the $i$th aerosol sample, $C_{i,j}$ is the average number concentration of the corresponding aerosol sample in the $j$th size range, and $d_j$ is the diameter of the particulates in the $j$th size range. Although the smallest detectable diameter of the particle counter was 0.3 μm, we consider the estimated specific areas to represent the areas associated with P in the samples because P is a substance in primary particles. The particle mass loading was estimated from the total aerosol Al concentrations by assuming that all aerosol Al was derived from mineral dust, which comprised 8 % of Al by mass (Taylor, 1964). In cases when the samples contained less mineral dust, the aerosol mass would be somewhat underestimated.

## 3 Results and Discussion

### 3.1 P concentration

The concentration of TP ranged from 24.7 to 392.6 ng m$^{-3}$, and the mean TP concentration was 125.5 ± 59.8 ng m$^{-3}$ (Fig. 1, Table 1). The highest and second highest values were observed in the samples on March 9 and April 8, when dust weather occurred and the concentration of both total Fe and total Al, the representative elements of mineral dust, exceeded 10000 ng m$^{-3}$. The TDP concentration ranged from 5.1 to 114.6 ng m$^{-3}$, and the mean TDP concentration was 32.4 ± 23.2 ng m$^{-3}$. Unlike TP, the highest concentration of TDP did not appear in the two dust samples and was rather measured in the samples collected during the haze and foggy days, mostly in January.





The DIP concentration in the TDP ranged from 1.5 to 102.6 ng m$^{-3}$, with an average concentration of 21.8 ng m$^{-3}$, accounting for 8.5-98.7 % of the TDP (Fig. 2). The DOP concentration in the TDP ranged from below the detection limit to 41.0 ng m$^{-3}$, with an average concentration of 10.7 ng m$^{-3}$, accounting for 1.3-91.5 % of the TDP. On average, approximately 60 % of the TDP in the aerosols was DIP. The DOP contribution to the TDP was not neglectable, with an average contribution of approximately 40 % of the TDP.

Table 1 also shows the results of previous studies on coastal and marine aerosol P concentrations. The TP concentration we observed in this study was lower than that observed in Singapore in spring (He et al., 2011) but slightly higher than the values obtained at Huaniao Island in the East China Sea (Guo et al., 2014) and higher than those observed at coastal sites of Spain and USA (Izquierdo et al., 2012; Zamora et al., 2013). The TP concentration at Qingdao was approximately one order of magnitude higher than that in the marine atmosphere (Baker et al., 2006b; Hsu et al., 2014; Sun et al., 2015). The DIP concentration in this study was obviously higher than those in the South China Sea, the Gulf of Aqaba and Miami, USA (Hsu et al., 2014; Chen et al., 2007; Zamora et al., 2013) but was lower than the values reported in Singapore and the Indian Ocean (He et al., 2011; Srinivas and Sarin, 2012). In fact, DOP was not measured in those studies (Table 1). Chen et al. (2007) studied DOP in aerosols over the Gulf of Aqaba and reported a DOP concentration and contribution of DOP to TDP that were similar to the results of this study.

There was a difference in the concentrations of total and dissolved P between the winter and spring (Fig. 1, Fig. 2). The average TP concentration in the spring was 128.6 ± 73.2 ng m$^{-3}$, slightly higher than that in the winter, i.e., 122.9 ± 45.6 ng m$^{-3}$. However, there was no statistically significant difference between the TP values of the two seasons ($p > 0.05$). Guo et al. (2014) also found that the atmospheric TP concentration in spring was higher than that in other seasons at Huaniao Island in the East China Sea. In contrast, over the Gulf of Aqaba, a higher aerosol TP concentration occurred in winter than in other seasons, although there were no statistically significant seasonal variations in TP (Anderson et al., 2010).

In contrast to the small seasonal difference in TP, TDP was higher in winter, with an average concentration of 40.5 ± 27.7 ng m$^{-3}$, and lower in spring, with an average concentration of 23.0 ± 11.0 ng m$^{-3}$. In TDP, the DIP in winter averaged 31.9 ± 23.4 ng m$^{-3}$, significantly higher than that in spring, which had an average of 10.0 ± 10.2 ng m$^{-3}$; in contrast, DOP in spring, with an average concentration of 13.0 ± 7.5 ng m$^{-3}$, was higher than that in winter, with an average concentration of 8.6 ± 7.4 ng m$^{-3}$. Furthermore, the ratio of DOP to TDP in spring was approximately 61 %, which was significantly higher than that in winter, i.e., 23 %. This result was probably caused by the release of primary biological particles and agriculture fertilization in spring (Mahowald et al., 2008; Kanakidou et al., 2012; Paytan and McLaughlin, 2007). Chen et al. (2006) also reported that, among all seasons, aerosols in spring had the largest percentage of organic P loads, a result potentially associated with terrestrial and marine biological activity.

TP was closely correlated with Al, Fe, Ca, Mn and Ba (Table S1). This is consistent with the fact that mineral particles from surface soils are the main origin of P. TP was also correlated with heavy metals of Cd, As, Ni, Zn and Pb, and with K. These results again support that the TP was derived from natural mineral dust, anthropogenic particles, biogenic particles, and emissions from biomass burning. Moreover, the correlation coefficients between TP and crustal elements (Al, Fe, Ca, Mn and





Ba) were larger in spring than in winter, while the close correlation between the TP and anthropogenic elements was observed in winter. This result suggests that a large proportion of TP in the winter was from anthropogenic sources. In contrast, the TP in the spring aerosols was substantially from crustal origins.

The DIP fraction in the TP also showed a significant correlation with Cd, As, Ni, Zn, Pb, K, non-sea salt (nss)-$SO_4^{2-}$ and nss-$K^+$ but had a miniscule correlation with Al, Fe, Ca and Ba (Table S1). This suggests that the DIP mainly originated from anthropogenic sources (particularly fossil fuel combustion and biomass burning) and biogenic sources (Weinberger et al., 2016; Zamora et al., 2013). Aerosol DOP is considered to be mainly derived from primary biological particles and biomass burning (Chen et al., 2006), which is supported by the correlation between DOP and K and nss-$K^+$ in the samples. We found that the DOP had a significant correlation with crustal source indicators (Al, Fe, Mn, Ba) and anthropogenic source indicators (Cd, As, Zn, Pb). Therefore, soil dust and anthropogenic emissions are potential sources of atmospheric DOP. Myriokefalitakis et al. (2016) reported that, at the global scale, approximately 50 % of atmospheric DOP is from primary biological aerosol particles, and the contributions of soil dust, anthropogenic combustion and biomass burning sources to DOP are approximately 25 %, 15.6 % and 9.4 %, respectively.

### 3.2 P Solubility

P solubility ranged from 7.3 % to 69.8 % (Fig. 1). Two low solubility values occurred in the dust samples due to the high concentration of TP and low concentration of TDP. High values occurred in the haze and foggy day samples, mostly in January. The average P solubility was 27.5 %, which is in agreement with the values reported at various coastal sites (Table 1). Anderson et al. (2010) reported that 15-30 % of P in Gulf of Aqaba aerosols was soluble. Guo et al. (2014) reported that the solubility of P was 0.6-63 % in the aerosols collected at the Huaniao Island of the East China Sea, and the median value was 21 % in spring. He et al. (2011) studied the atmospheric dry and wet deposition of P in Singapore and reported that DIP comprised 38 % of TP in aerosols. In the dust samples of this study, however, the P solubility was approximately 8 %, which was close to the value of 10 % measured in dust aerosols collected at the Mediterranean coast (Carbo et al., 2005; Herut et al., 2002) but was considerably lower than the value of 20 % measured in Asian dust at the coastal site of the East China Sea and in African dust in the western subtropical North Atlantic (Guo et al., 2014; Zamora et al., 2013). There are few available data for P solubility in the marine atmosphere (Hsu et al., 2014; Baker et al., 2006a, b), and the limited data show that the solubility of P in marine aerosols is generally higher than that in coastal aerosols (Table 1).

The ratio of DIP/TP has frequently been used as a proxy for P solubility in previous studies with no consideration of the contribution of DOP. The fraction of DOP in TDP was approximately 40 % in this study (Fig. 2) and 31 % in the aerosols at the Gulf of Aqaba (Chen et al., 2007). Our DOP results indicate that using the DIP/TP ratio as a proxy for the solubility of P could largely underestimate the real P solubility in aerosols by approximately 30-40 %.

Seasonally, the P solubility in winter was 32.9 ± 16.7 %, which was much higher than the value of 21.3 ± 9.8 % for P solubility in spring (Fig. 1, Table 1). In winter, the P solubility was negatively correlated with Al (Table S1) and positively correlated with anthropogenic elements such as Zn, As, Cd, Pb and ions such as nss-$K^+$, nss-$SO_4^{2-}$ and $NO_3^-$. The correlations



likely result from the effects of crustal and anthropogenic sources, as well as the effects of atmospheric processes on P solubility. However, in spring, P solubility did not have a clear relationship with anthropogenic source indicators, i.e., Cd, As, Zn, Pb, nss-$SO_4^{2-}$ and $NO_3^-$, but was still negatively correlated with crustal elements such as Al, Fe, Ca, Mn and Ba. Therefore, the variation in P solubility could also be caused by changes in the dominant aerosol particles.

**3.3 Factors influencing P solubility**

**3.3.1 Aerosol sources and mineral dust loading**

Mineral dust aerosols are generally considered the dominant sources of atmospheric TP (Mahowald et al., 2008). As mentioned above, TP had statistically significant correlations with the contents of mineral elements. We choose Al as the indicator of the mineral origin of aerosols to investigate the effect of aerosol sources and dust loading on P solubility. Because the content of Al in mineral particles is stable, Al has few sources other than mineral particles (Arimoto et al., 2006). TP showed a strong linear relationship with total Al (Fig. 3a), and all data points were significantly above the line of P/Al = 0.013, which is the ratio of P to Al in the crust (Taylor, 1964). Therefore, there were sources of TP other than the crustal source.

The concentration of anthropogenic P ($P_{anth}$) was estimated with the following formula (2):

$$P_{anth} = TP - TAl \cdot \frac{P_{crust}}{Al_{crust}} \tag{2}$$

where TP is the total P concentration, TAl is the total Al concentration, and $P_{crust}$ / $Al_{crust}$ is the ratio of P to Al in continental crust. In fact, the $P_{anth}$ from the above formula is non-dust P. We consider the excess P relative to P/Al in the mineral dust to represent the fraction of P derived from anthropogenic activities, because atmospheric P is mainly caused by anthropogenic activities and natural dust besides a small fraction of biological P. Results show that the maximum concentration of anthropogenic P was about 75.8 ng m$^{-3}$, as shown by the intercept of the regression line in Fig. 3a, and the maximum contribution of anthropogenic P to TP was as high as 60 % on average. Part of the TP was also from primary biological sources, especially in spring. Therefore, the value 60 % overestimates the P contribution of anthropogenic sources.

Aerosol P from anthropogenic sources usually has a higher solubility than that from mineral sources. The reason is that anthropogenic P tends to loosely associate with particulates and dissolve more readily than mineral P and, consequently, easily interact with pollutant acid gases to produce more bioavailable P (Herut et al., 2002; Baker et al., 2006a; 2006b; Anderson et al., 2010; Hsu et al., 2014; Herbert et al., 2018). Our results indicate that the P solubility was approximately 35 % in the aerosols when the fraction of anthropogenic P in the TP was more than 70 %. In contrast, the value was approximately 15 % in the aerosols when the fraction of anthropogenic P in the TP was less than 50 %. Therefore, the high fraction P from anthropogenic sources in the TP in winter is likely one reason that winter P solubility was higher than spring P solubility.

Similar to the correlation between TP and total Al, there was a high correlation between TDP and soluble Al (Fig. 3b). The slope of the regression line between P and Al was 0.13 in the soluble fraction, which was higher than that in the total fraction. This result is reasonable because the solubility of P is considerably higher than that of Al (Hsu et al., 2014) and/or because





anthropogenic sources directly contribute to more soluble-fraction P (Anderson et al., 2010).

The P solubility and total Al concentration (as dust loading) displayed an inverse power-law relationship (Fig. 4). The data in the China Sea from the literature also conforms to this fitting relationship (Hsu et al., 2014; Guo et al., 2014), except for the data observed over the East China Sea in July and August 2010, when the air mass was mainly from the open ocean (Guo et al., 2014). A characteristic inverse relationship with Al has also been observed for aerosol Fe over large regions of the global ocean (Sholkovitz et al., 2012). Similarly, the relationship between P and Al can be attributed to the mixing of two end-members, i.e., mineral dust with a low P solubility and high Al loading and anthropogenic aerosols with a high P solubility and low Al loading.

For the samples with high Al loading, i.e., a concentration of more than 6000 ng m$^{-3}$, the P solubility was frequently below 15 %. The TP in these samples was mainly derived from mineral dust, which contributed more than 60 % of the TP. The 72 h back trajectories for samples with high Al loading indicate that the air masses originated from the arid and semiarid areas in the inland Asian continent, where aerosol particles are dominated by natural mineral dust (Fig. S2). Moreover, the air masses were rapidly transported in the elevated layer; the average highest altitude was approximately 2,600 m during the transport of these air masses, and the average transport speed in the 36 h before arrival at Qingdao was 47 km h$^{-1}$. These air masses passed populated areas in a short period of time and experienced little interferences by anthropogenic pollutants; therefore, the high Al loading and low P solubility end-member exhibited a relatively high mineral dust characteristic.

For samples with a low Al loading below 2000 ng m$^{-3}$, the P solubility was usually above 25 %, and the average P solubility was approximately 35 %. The TP in these samples was mainly from anthropogenic sources, which contributed approximately 70 % of TP. The 72 h back trajectories for the samples indicated that, very different from the air masses with high Al loading, these air masses moved slowly at a low altitude (Fig. S3). The highest altitude during the air mass transport was approximately 1,300 m on average, and the transport speed in the 36 h prior to arrival at Qingdao was an average of 23 km h$^{-1}$. The air masses passed populated areas and were influenced directly by anthropogenic pollutants. Therefore, the low Al loading and high P solubility end-member exhibited a relatively high concentration of anthropogenic aerosols.

The inverse relationship between P solubility and total Al loading may also reflect the effects of particle size, similar to aerosol Fe (Baker and Jickells, 2006; Baker and Croot, 2010). A dust population with a small modal size has a large surface area, which more efficiently serves as a sink for acidic constituents and easily leads to an increase in the solubility of dust-derived elements. There was a clear linear correlation between the P solubility and the specific surface areas of the aerosols, and the solubility increased with decreases in the particle size (Fig. 5). This result is consistent with previously reported results on this subject (Baker et al. 2006a; b).

### 3.3.2 Atmospheric acidification processing

Some data points deviated from the fitting curves (Fig. 4, Fig. 5), and most of these points presented in the samples containing more anthropogenic particles. This result suggests that the two end-member mixtures of mineral dust and anthropogenic particles could not completely explain the P solubility. Nenes et al. (2011) proposed that the atmospheric acidification



processes that mineral aerosols experience could be a primary mechanism to enhance aerosol P solubility. Nenes et al. found that the solubility of P in the Sahara surface soil and dust aerosols increased by 10-40 times after acid treatment using a pH 2 sulfuric acid solution. Stockdale et al. (2016) also found that the amount of P dissolved is directly proportional to the amount of $H^+$ consumed at $H^+ > 10^{-4}$ mol g$^{-1}$ of dust. Similar processes have been demonstrated to affect the fraction of soluble Fe in

mineral dust (e.g., Baker and Croot, 2010; Shi et al., 2015; Longo et al., 2016; Li et al, 2017). However, a corroboration of direct observational data on aerosol P solubility remains insufficient. Hsu et al. (2010) reported that soluble P correlated well with nss-$SO_4^{2-}$ and $NO_3^-$ in aerosols collected over the East China Sea and implied that the dissolution of aerosol P was enhanced by the presence of acidic constituents. In the present study, we confirmed that both soluble P and P solubility were statistically correlated with nss-$SO_4^{2-}$, $NO_3^-$ and $Cl^-$, indicating the potential enhancement of the dissolution of aerosol P by

acid processing and the consequent increase in P solubility.

    Because the ratio of acids/total Fe has been employed to demonstrate the influence of degree of aerosol acidification on aerosol Fe solubility (Hsu et al., 2014), here, we used the ratio of acids/total P to demonstrate the influence of degree of aerosol acidification on aerosol P solubility. We found that both the nss-$SO_4^{2-}$/TP and $NO_3^-$/TP molar ratios were statistically correlated with P solubility in all samples ($r = 0.36$, $p < 0.01$) (Fig. 6a, 6b). In contrast, there was no such relationship between the $Cl^-$

/TP molar ratio and P solubility (Fig. 6c). The combination of nss-$SO_4^{2-}$ and $NO_3^-$, the two major acidic constituents in atmospheric aerosols, was further examined (Fig. 6d). The samples were classified into three groups based on the relative contribution of anthropogenic P to TP ($P_{anth}$/TP) (Fig. 6). The P solubility increases with the acidification degree (the molar ratio of [2nss-$SO_4^{2-}$+$NO_3^-$]/TP) of aerosols. However, the P solubility versus the acid/TP ratio followed different regression curves corresponding to the ranges of $P_{anth}$/TP. The slope of the regression curve was greater in samples with $P_{anth}$/TP > 70 %

than in samples with $P_{anth}$/TP < 50 %. The aerosol specific surface area in the samples with $P_{anth}$/TP > 70 % was 4.54 m$^2$ g$^{-1}$, three times larger than that of the samples with $P_{anth}$/TP < 50 %, i.e., 1.45 m$^2$ g$^{-1}$. The samples with $P_{anth}$/TP > 70 % had more areas to react with acidic constituents, leading to more TDP. However, this explanation can only explain approximately 15 % of the difference between the slopes of the two fitted curves. The remaining difference should be due to the different natures of P in the aerosols; therefore, P in anthropogenic particles should be more susceptible to acidic processing than that in natural

mineral dust. Notably, the anthropogenic P in the aerosols had an intrinsically higher solubility than the soil-derived mineral P (Baker et al., 2006a; 2006b). This means that changes in the dominant aerosol particles could also cause differences between the slopes of the two fitting curves. Hsu et al. (2014) also found that the relationship of aerosol P solubility vs. ∑(nss-sulfate+nitrate)/TP in a February-March cruise and June cruise over the South China Sea followed two regression curves. They attributed the discrepancy to the discernible dominance of the respective sources vs. the susceptibility to acidic processes.

Two data points largely deviated from the fitting curve of the $P_{anth}$/TP > 70 % samples (Fig. 6). Analysis of the characteristics of the two samples revealed that their DOP content was relatively high and was approximately 21 % of the TP, while the DOP value for other samples was 7 % of the TP. Aerosol organic P is originally associated with organic matter in biological sources rather than anthropogenic sources (Chen et al., 2006). Therefore, the contribution of anthropogenic P to TP was significantly overestimated in the two samples due to the high organic P content. In addition, insoluble organic P could be converted to



DOP due to the uptake of oxidants and the formation of large chains of soluble multifunctional groups (Ariya et al., 2009; Myriokefalitakis et al., 2016). The conversion mechanism of organic P from insoluble to soluble, which is different from the acid-solubilization of P, may also be responsible for the deviation of the two samples from the fitted curve.

### 3.3.3 Relative humidity

The data points for the samples of 50 % < $P_{anth}$/TP < 70 % were frequently between the two fitted curves of $P_{anth}$/TP > 70 % and $P_{anth}$/TP < 50 % (Fig. 6) and had a statistically significant correlation at the 99 % confidence level ($r = 0.383$, $p = 0.006$). Some data points largely deviated from the fitted curves, showing high P solubility at low degrees of acidification. We focused on the seven deviating data points at P solubility > 45 % and acid/TP < 200 (the $P_{anth}$/TP in the samples was approximately 60 %, and these data points are highlighted in red circles in Fig. 6). The relative humidity during the collection of these samples

was approximately 80 % on average, much higher than the value of 65 % for the other samples in this group. Elevated RH can result in particle phase conversion from a semisolid to a liquid state (Liu et al., 2017). The presence of aqueous layers on the aerosol particles can enhance insoluble trace elements, such as Fe, to dissolve under acidic conditions (Shi et al., 2015). This was likely caused by the acidification of particles as they cycled from cloud droplets to wet aerosols and back (Nenes et al., 2010; Stockdale et al., 2016). We further investigated the relationship of RH with P solubility and confirmed that the two

factors showed a significant correlation ($r = 0.62$, $p < 0.01$), suggesting the significance of RH as one of the factors influencing aerosol P solubility.

The coordinated effect of RH, aerosol sources and acidity on P solubility was further investigated (Fig. 7). Because the threshold of RH for particle conversion from the semisolid to liquid state was frequently approximately 60 % (Liu et al., 2017), we used 60 % RH as the threshold of wet aerosol formation. The P solubility was frequently lower than 30 % when RH <

60 %, even in samples with a relatively high acidification degree or a large fraction of anthropogenic P. On average, the P solubility was approximately 13 % in the aerosols of $P_{anth}$/TP < 50 %, while the value was approximately 21 % in the aerosols of $P_{anth}$/TP > 50 %. The aerosol acidification had little effect on the P solubility when the RH was less than 60 %.

At RH > 60 %, the P solubility ranged from approximately 10 % to approaching 70 %. To examine the influence of acidity or humidity on P solubility, the samples were classified into two groups according to the acidification degree of 150 nmol

nmol$^{-1}$, which was close to the average acidification degree of all aerosol samples. For the aerosols of $P_{anth}$/TP < 50 %, the P solubility was higher than 13 %. Under similar acidic conditions, P solubility increased with the increase in RH. However, the increase was not significant and was only approximately 2 % when the RH changed from < 60 % to > 60 %. For the aerosols of 50 % < $P_{anth}$/TP < 70 %, the P solubility was frequently higher than 21 %. P solubility increased with the increase in RH. When RH increased from less than 60 % to more than 60 %, the P solubility increased from, on average, 20 % to 30 % under

the acid/TP < 150 condition and approximately 20 % to 35 % under the acid/TP > 150 condition. For the aerosols of $P_{anth}$/TP > 70 %, the P solubility was higher than 21 %. Under similar acid conditions, the P solubility was in the range of 40 % -60 % at RH > 80 % and in the range of 20 % -40 % at 60 % < RH < 80 %.

Comparing the above results, we found that the increase in P solubility with the increase in RH was significantly higher in



the anthropogenic aerosols than in the mineral aerosols, suggesting again that P from anthropogenic sources was more susceptible to acidification than P from mineral sources. All of these results indicate that RH is also likely one of the factors significantly impacting aerosol P solubility; high humidity could facilitate the dissolution of aerosol P under acidic conditions and hence increase its solubility.

## 4 Conclusions

In this study, we quantified the levels and seasonal variability of TP, TDP, DIP, DOP and P solubility at Qingdao, a coastal city of the Yellow Sea, China, aiming to demonstrate the solubility of P in aerosol particles from Asian continental outflow. The P solubility in Qingdao aerosols was 7.3-69.8 %, which overlapped with the range of values reported in other coastal areas. P solubility in winter was significantly higher than that in spring, which was related to the aerosol sources, atmospheric acidification processes and ambient relative humidity. Our data indicated that the P in aerosols from anthropogenic sources had a higher solubility than the P in aerosols from mineral sources. The acid processing associated with sulfate and nitrate formation could increase the solubility of P from mineral dust or anthropogenic sources, with P from anthropogenic sources demonstrating a greater susceptibility to acid processing than P from mineral dust. In addition, we found that DOP has an important contribution to dissolved P, with an average contribution of 40 % in some cases. Furthermore, ambient relative humidity was an important factor influencing P solubility. At RH < 60 %, it was difficult for P solubility to exceed 30 %, even with a high aerosol acidity and anthropogenic P contribution. High RH levels increased the dissolution of aerosol P to a greater degree under acidic conditions, consequently increasing P solubility. The threshold RH for this effect was approximately 60 %. These results imply and support that anthropogenic inputs could provide more soluble P to the ocean via the atmosphere, thus increasing the bioavailable P in the ocean and potentially impacting ocean biogeochemistry.

*Acknowledgement.* We appreciate Xiaoyu Ben for his assistance in sample collection and part chemical analysis. This research received support from the National Key Research and Development Program of China (2016YFC0200504), the National Key Basic Research Program of China (2014CB9537001) and the National Nature Science Foundation of China (41876131).

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





**Table 1 Comparison of the concentrations of various forms of P and P solubility in aerosols from the coastal Yellow Sea (this study) and other previously studied coastal and marine sites.**

| Region | Period | Type | TP (ng m$^{-3}$) | DIP (ng m$^{-3}$) | DOP (ng m$^{-3}$) | P solubility (%) | Reference |
|---|---|---|---|---|---|---|---|
| Coastal site | | | | | | | |
| Qingdao,China | December 2012- April 2013 | TSP | 125.5±59.8 | 21.8±21.4 | 10.7±7.7 | 27.5±15.0 | This work |
| | December 2012-January 2013 | TSP | 122.9±45.6 | 31.9±23.4 | 8.6±7.4 | 32.9±16.7 | |
| | March -April 2013 | TSP | 128.6±73.2 | 10.0±10.2 | 13.0±7.5 | 21.3±9.8 | |
| Huaniao Island, East China Sea | April 2-30, 2010 | TSP | 15-307 | | | 0.6-58 | Guo et al., 2014 |
| | July 29-August 27, 2010 | TSP | 14-115 | | | 6.9-14 | |
| | November 12-December 11, 2010 | TSP | 14-159 | | | 1.8-63 | |
| | March 4-30, 2011 | TSP | 2.6-113 | | | 4.5-56 | |
| Singapore | April 2007-March 2008 | TSP | 430±320 | 120±120 | | 38 | He et al., 2011 |
| Mediterranean coast | March 2002-December 2003 | TSP | 22.9±1.1 | | | | Izquierdo et al., 2012 |
| Miami, Florida, USA | January 2007-August 2008 | Bulk aerosols | 11.8±10.8 | 3.1±2.8 | | 23 | Zamora et al., 2013 |
| Barbados, West Indies, USA | July 1988–September 2008 | Bulk aerosols | 11.8±9.7 | 2.6±1.3 | | 21 | |
| Gulf of Aqaba | August 2003-November 2004 | TSP | | | | 15-30 | Anderson et al, 2010 |
| Gulf of Aqaba | August 2003-September 2005 | TSP | | 12.4±6.2 | 6.2±3.1 | | Chen et al., 2007 |
| Marine site | | | | | | | |
| East China Sea and Japan Sea | June-September 2008 | TSP | 7.9±6.4 | | | | Sun et al., 2015 |
| South China Sea | February 18-March 8, 2013 | TSP | 10±11 | 5.6±4.7 | | 55±14 | Hsu et al., 2014 |
| | June 17-30, 2013 | TSP | 36±22 | 16±12 | | 45±17 | |
| Western North Pacific | June-September 2008 | TSP | 6.9±6.6 | | | | Sun et al., 2015 |
| Atlantic Ocean | September 10-October 24, 2001 | Bulk aerosols | 0.1-5.6 | | | 0.01-87 | Baker et al., 2006b |
| Indian Ocean | January 2009 | PM$_{10}$ | | 40.3±15.5 | | | Srinivas and Sarin., 2012 |
| | March-April 2006 | PM$_{10}$ | | 34.1±12.4 | | | |
| Arctic Ocean | June-September 2008 | TSP | 7.0±6.8 | | | | Sun et al., 2015 |





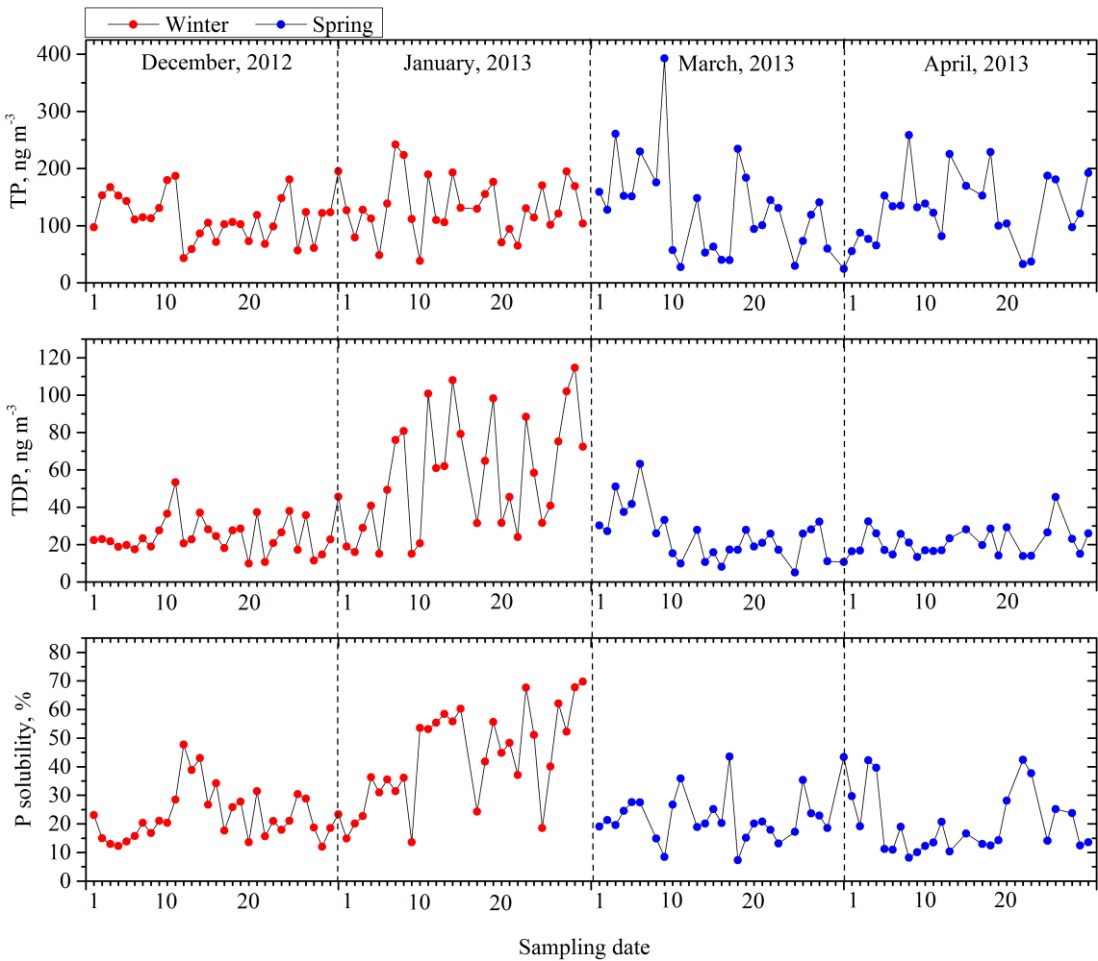

**Figure 1: Time series of total and dissolved P concentrations (TP and TDP) and P solubility.**





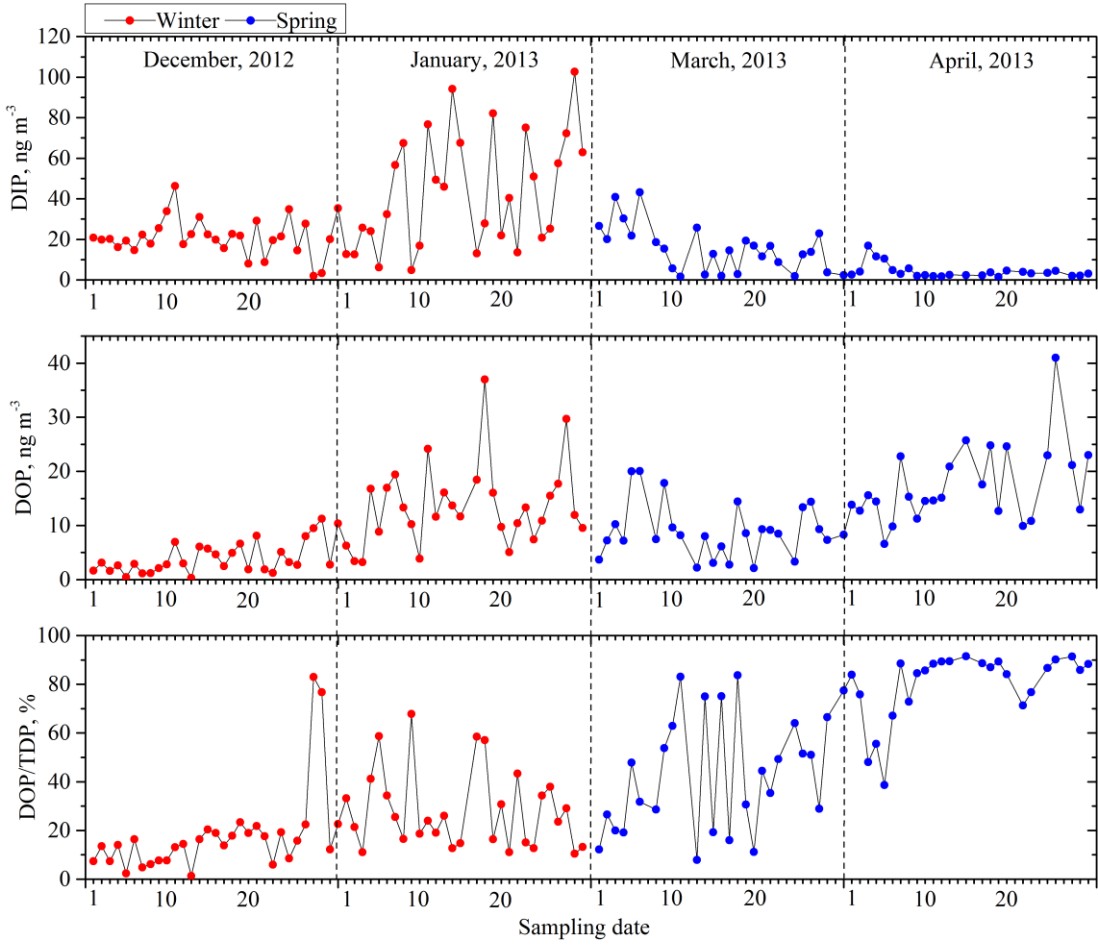

**Figure 2: Time series of dissolved inorganic and organic P concentrations (DIP and DOP) and the percentage of DOP in TDP.**



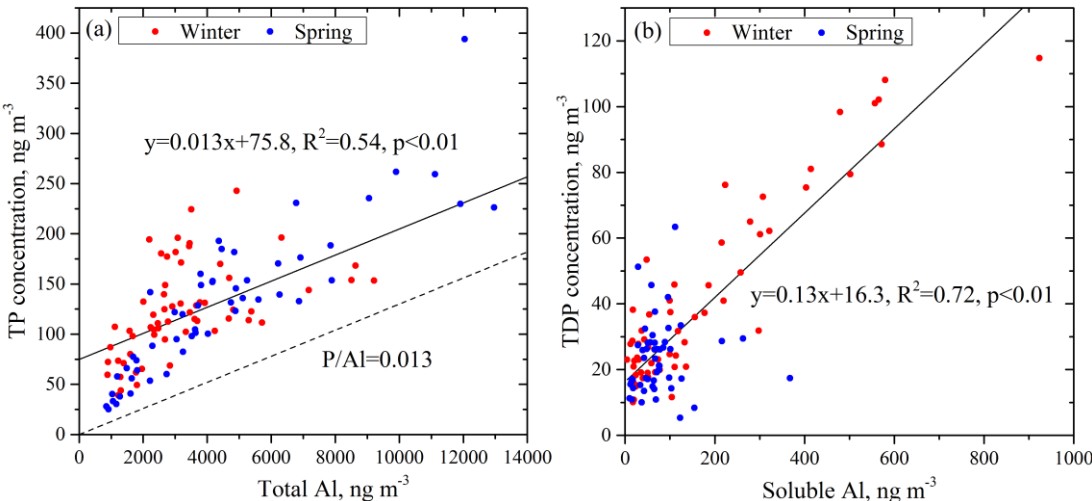

**Figure 3: Correlations of TP with total Al (a) and TDP with soluble Al (b). The line of P/Al = 0.013 represents the ratio of P to Al in the crust.**





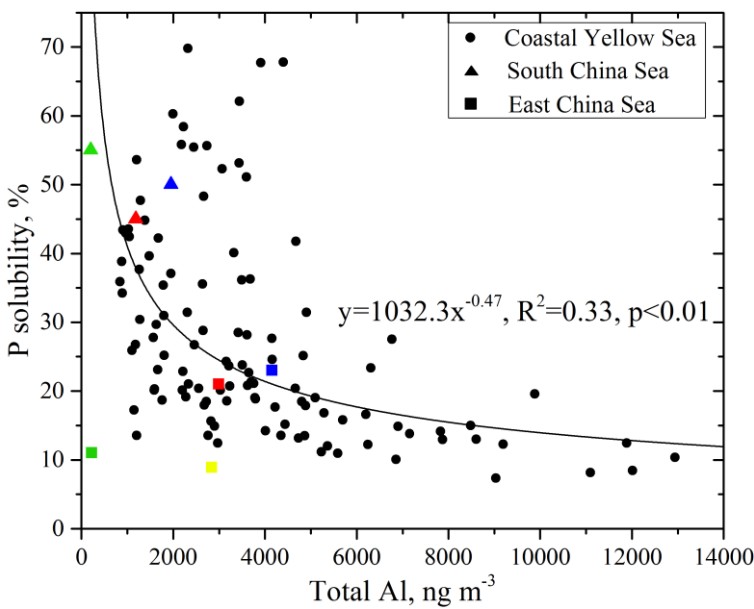

**Figure 4: Relationships of P solubility against total Al. Data in the South China Sea are from Hsu et al. (2014) (triangles), and data in the East China Sea are from Guo et al. (2014) (squares). Different color symbols are used to highlight different observation periods. The green and red triangles represent the averages of observations in the South China Sea during cruises in February-March 2013 and June 2013, respectively; the blue triangles represent the observed values of the samples affected by biomass burning in the June cruise. The blue, green, yellow and red squares represent the averages of observations over Huaniao Island in the East China Sea during April, July-August and November-December 2010 and March 2011, respectively.**


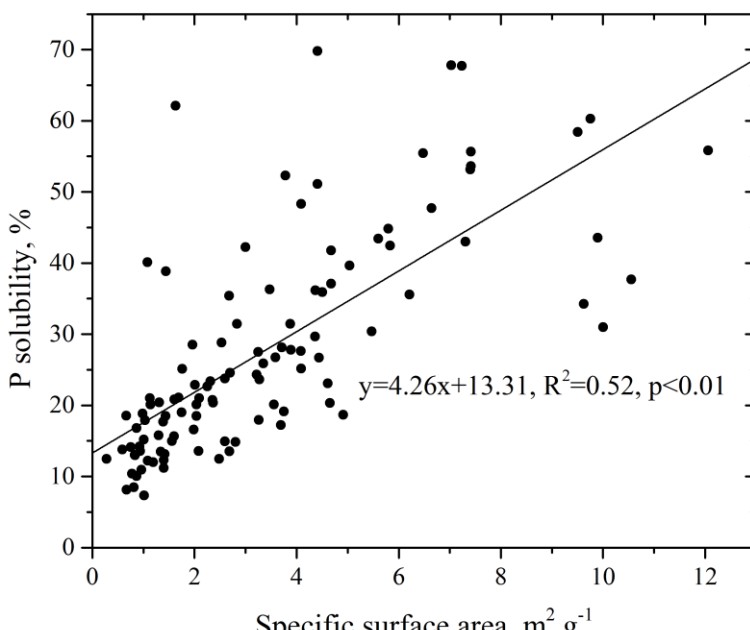

**Figure 5: Relationships between P solubility and the specific surface area of particles.**





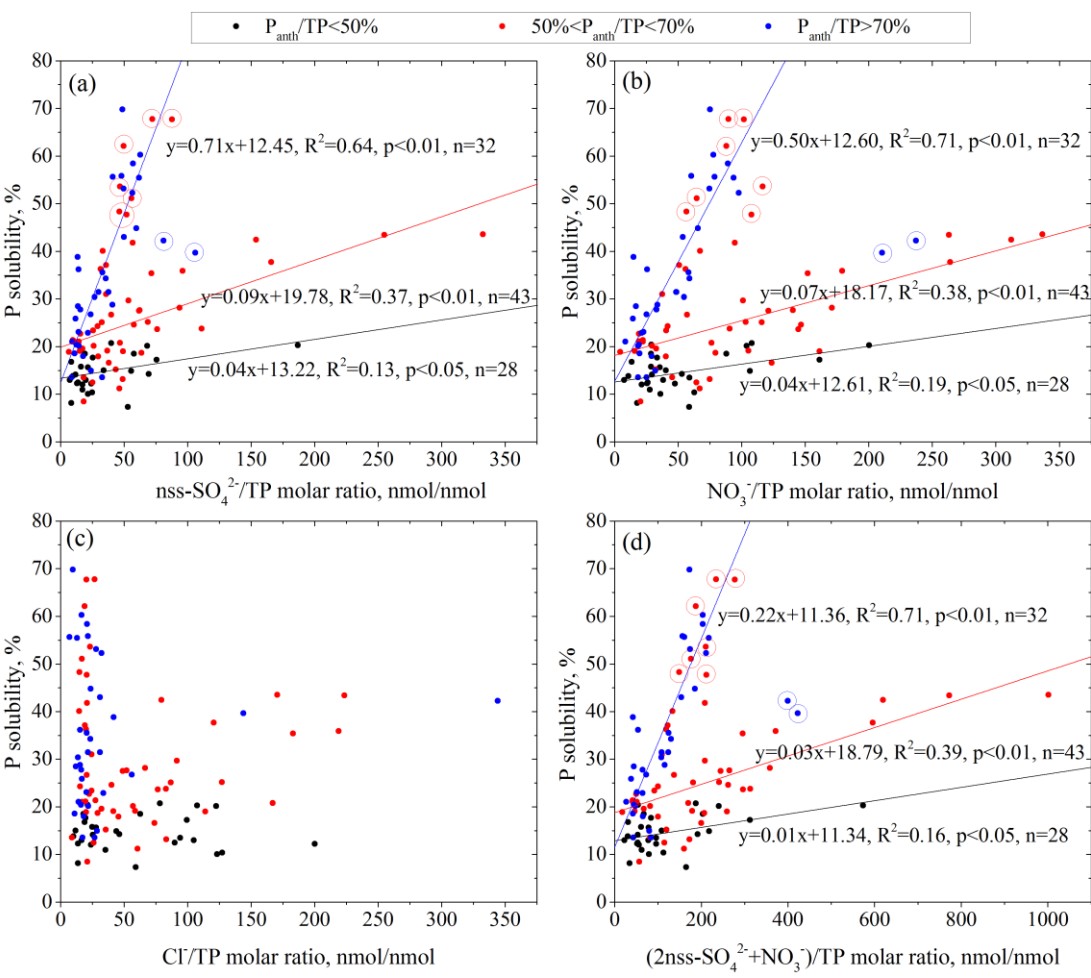

**Figure 6: Correlations between P solubility and the nss-SO$_4^{2-}$/TP molar ratio (a), NO$_3^-$/TP molar ratio (b), Cl$^-$/TP molar ratio (c) and (nss-SO$_4^{2-}$+NO$_3^-$)/TP neq/molar ratio (d) in aerosols with different relative contributions of anthropogenic P to TP (P$_{anth}$/TP). Data points within the circle were not included in the regression fitting.**





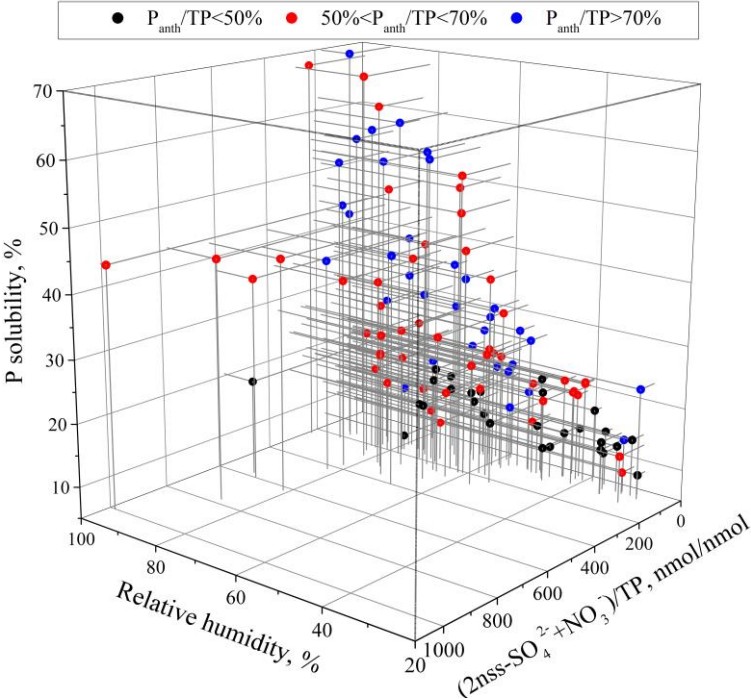

**Figure 7: Relationships of P solubility with the relative humidity and acidification degree in aerosols with different ranges of $P_{anth}/TP$.**