# Peer review of "Phosphorus solubility in aerosol particles related to particle sources and atmospheric acidification in Asian continental outflow"

_Atmospheric Chemistry and Physics, 2018_

## Referee Comment (RC1) · Anonymous Referee #1 · 18 Oct 2018

General Comments: I applaud the aim of this manuscript and I feel once it is modified slightly, that it will make an important contribution to the literature. I should say that a couple of years ago, we tried to do exactly the same data treatment using a data set collected in Crete. We had in total ~100 data points and we were unable to find significant patterns. This manuscript has 170 data points and has managed (just) to see some real patterns albeit the correlations they find are often statistically significant but with correlation coefficients of ~0.3!. In other general words, while the conclusions are interesting, they are not actually very strong. In particular the authors seem to divide the particles into anthropogenic or dust only. They do not include the importance of acid processing of inorganic particles (dust or anthropogenic) as wet aerosols asso-

ciated with clouds (high relative humidity) as a potentially important process. In fact they do discuss this in the text and state on page 10 that "Unfortunately , we were unable to quantitively distinguish the contributions of aerosol source and acidification to phosphorus solubility at this stage." Yet the text elsewhere minimises the possible contribution of acidification and emphasises instead anthropogenic particles which had high P solubility at source. This reviewer feels the manuscript would benefit by taking a more even balance between these two possibilities. As a final general point, we have just published a paper in Global Biogeochemical Cycles (Herbert et al., 2018), which the authors obviously could not have seen. However it does predict that acid processes in China could be an important source of. Bioavailable P as a plume which passes over location such as Qingdoa and on to the western Pacific.

Herbert, R. J., Krom, M.D., Carslaw, K.S., Stockdale, A., Mortimer, R.J.G., Benning, L.G., Pringle, K., Browse, J., (2018) Quantifying the effect of atmospheric acid processing on the global deposition of bioavailable phosphorus from dust. Global Biogeochemical Cycles. (5.79) https://doi.org/10.1029/2018GB005880

Specific comments: Line 16 of Abstract and elsewhere: The convention for what is called in this manuscript DP, is actually TDP (Total Dissolved P). That is the P measured after persulphate oxidation in solution. Please change to TDP throughout.

Line 24: The authors suggest that humidity plays an important role in converting refractory P to bioavailable P. The most likely mechanism is that suggested in Nenes et al., (2010) which is the acidification of particles as they cycle from clouds, where the pH is rather high, to wet aerosols (where the pH is very low) and back again (see Stockdale et al., (2016).

Introduction Line 5 Add in offshore areas and regions where P limits. . ...

General: Even in systems where N is the immediate limiting nutrient P can increase phytoplankton growth by moving the entire system to higher productvitiy.

Introduction page 2 line 9: The authors should comment/introduce the idea that anthropogenic processes can include the production of atmospheric acids, which can cause previously unreactive p to become bioavailable DIP. They discuss this possibility at length towards the end of their manuscript.

Methods page 4 line 7 Remove 'in number of particles' and Replace monitored with measured.

Page 5 line 19: What was the assumed value of Al in mineral dust that allowed the authors to assume that the particles were 8% by mass? I may have misunderstood what was written, in which case the authors should make it clearer.

Page 5 line 25 What is 'floating' dust? A dust storm?

Page 6 line 11 (and various other places including table 1) Aqaba is spelt wrongly. It is a b and not a d

Page 6 lines 31-34: If TP had high correlations with major elements (dust) and with heavy metals (anthropogenic) at the same time, is that not ambiguous?

Page 7 line 3: The actual correlation data is not given (or at least not given here). This reviewer is a little confused as to what the authors mean by 'higher correlations' and whether that also means lower p values.

Page 7 line 13 Are the authors convinced that soil dust (from deserts?) are an important source of DOP?

Page 7 Line 25 The authors might consider quoting Carbo et al., (2005) which presents the P solubility data for the Eastern Mediterranean in a more comprehensive manner than Herut et al. (2002). Carbo, P., Krom, M.D., Homoky, W.B., Benning, L.G., Herut, B., 2005. Impact of atmospheric deposition on N and P geochemistry in the southeastern Levantine basin. Deep-Sea Research II Volume 52: Nos 22-23, 3041-3053.

Page 8 line 24: The data in that graph is non-linear

[Figure]

Page 8 line 27 And because anthropogenic P is more likely to have interacted with pollutant gases to produce more bioavailable P

Page 9 line 4 Remove obviously

Page 9 line 14 I had the same problem with Sholkovitz's paper too. It ignores the possibility that anthropogenic acids can interact with mineral dust to produce bioavailable P (or Fe). The authors of this article suggest this might be an important process themselves in line 31 "which more efficiently serves as a sink . . ..derived elements." And later on page 10 "Unfortunately we were unable to quantitatively distinguish the contributions of aerosol source and acidification to phosphorus solubility at this stage". That means both should be retained as possible sources. In reality the answer is probably that both more soluble P in anthropogenic particles at source and more P made soluble by acid processes in air masses from polluted sources occur and are in different proportions in different air masses.

Page 11 line 29 There seems to be a mistake in the first half of the line. I read it several times and could not decide what was meant.

Page 11 line 34 How did the authors define 'acidification degree of 150 nmol nmol-1? nmoles of what?

Page 12 line 3 Remove obviously

Page 12 Conclusions Very well written and this reviewer entirely agrees with the conclusions.

---

## Referee Comment (RC2) · Anonymous Referee #2 · 26 Oct 2018

The manuscript investigated the P speciation and solubility in aerosols in the eastern China's coast. Phosphorus may ultimately control the primary production in the large areas of the ocean especially in the N-affluent regions such as the marginal seas of the western North Pacific. The previously reported P solubility in aerosols was in a wide range, and therefore it is important to understand the factors or mechanisms determining the atmospheric input of soluble P. The manuscript studied the coordinated effect of relative humidity (RH) and aerosol origins and acidity on P solubility, and also included dissolved organic P in the discussion. The manuscript indicated that P in aerosols from anthropogenic sources had a higher solubility than the P in aerosols from mineral dust. Phosphorus solubility was usually less than 30% when the RH

was below 60% and the higher RH increased the dissolution of aerosol P to a great degree under acidic conditions (how to define acidic condition?). These results will be very helpful to modeling the input of bioavailable P to the ocean. It would be nice if authors could discuss extrapolation of the results to the eastern China seas, other coastal regions or even the open ocean.

Introduction Page 2 Line 3: Authors may add few sentences on the importance of atmospheric P deposition to the surface ocean. For example, long-term measurements of dissolved P at station ALOHA revealed unexpected temporal variability in PO43- concentrations in the surface ocean, which may be partly due to the episodic atmospheric deposition (Karl and Tien 1997).

Karl DM,TienG. 1997. Temporal variability in dissolved phosphorus concentrations in the subtropical North Pacific Ocean. Mar. Chem. 56:77–96

Methods Page 4 Line 26, "P" at the beginning of the sentence should be changed to "Phosphorus"

Page 5 Line 18: "because P is a substance in primary particles". Here what is the general size range for primary particles? Authors may provide the reference or the size distributions of P and DP to prove the statement . Page 5 line 18-20: "In cases when the samples contained less mineral dust, the aerosol mass would be somewhat underestimated." The mass loadings estimated from Al concentrations may be compared to the officially reported PM10 concentrations to check for the average underestimation.

Results and Discussion Page 6 Line 3-5: The two sentences can be combined to be more concise.

Page 6 Line 27: "This result was probably caused by the release of primary biological particles and agriculture fertilization in spring." Was DOP released by the agricultural process in spring or as the loss of fertilizer (I thought that fertilizer should be mainly DIP)?

Page 8 Line 18: The statement "a small fraction of biological P" needs to be supported by a reference or observatory data.

Page 8 Line 29-31: Why is the correlation of TDP vs soluble Al better than that of TP vs Al? Such comparison is hard to explain. Authors may delete this sentence and just compare the ratios of P/Al to the ratios of TDP/soluble Al.

Page 9 Line 31: "Some data points deviated from the fitting curves." The specific variables for fitting curves should be indicated here, e.g. for P solubility and total Al.

Page 10 Line 11-13 & 18-20: There is repetition in these sentences.

Page 11 Line 6: "had a statistically significant correlation" What are the variables?

Page 11 Line 17-32: The coordinate effect could be arranged as another section. The relationships between P solubility and humidity, anthropogenic percentage and acidification are complex. The two paragraphs seem to talk about the situations at RH<60% and RH>60% respectively. But the RH change from <60% to >60% was discussed again in the second paragraph. This part needs to be reorganized.

Page 11 Line 25: The unit of acidification degree should be unified in the paper. Is it proper to choose 150 acidification degree as the boundary?

Page 12 Line 1-2: The first half sentence talked about the effect of RH on P solubility, and the second half mentioned acidification. The linkage between the RH and acidification was missing.

Conclusion Page 12 Line 12: "...from mineral dust and anthropogenic sources" can be deleted.

Page 12 Line 17: "The threshold RH for this effect was approximately 60%." can be deleted.

---

## Author Comment (AC1) · 13 Dec 2018

Responses to Reviewer 1# Comments

Reviewer #1 (Comments to Author):

General Comments: I applaud the aim of this manuscript and I feel once it is modified slightly, that it will make an important contribution to the literature. I should say that a couple of years ago, we tried to do exactly the same data treatment using a data set collected in Crete. We had in total ∼100 data points and we were unable to find significant patterns. This manuscript has 170 data points and has managed (just) to see

some real patterns albeit the correlations they find are often statistically significant but with correlation coefficients of ∼0.3!. In other general words, while the conclusions are interesting, they are not actually very strong. In particular the authors seem to divide the particles into anthropogenic or dust only. They do not include the importance of acid processing of inorganic particles (dust or anthropogenic) as wet aerosols associated with clouds (high relative humidity) as a potentially important process. In fact they do discuss this in the text and state on page 10 that "Unfortunately , we were unable to quantitively distinguish the contributions of aerosol source and acidification to phosphorus solubility at this stage." Yet the text elsewhere minimises the possible contribution of acidification and emphasises instead anthropogenic particles which had high P solubility at source. This reviewer feels the manuscript would benefit by taking a more even balance between these two possibilities. As a final general point, we have just published a paper in Global Biogeochemical Cycles (Herbert et al., 2018), which the authors obviously could not have seen. However it does predict that acid processes in China could be an important source of. Bioavailable P as a plume which passes over location such as Qingdoa and on to the western Pacific.

Herbert, Herbert R. J., Krom, M.D., Carslaw, K.S., Stockdale, A., Mortimer, R.J.G., Benning, L.G., Pringle, K., Browse, J., (2018) Quantifying the effect of atmospheric acid processing on the global deposition of bioavailable phosphorus from dust. Global Biogeochemical Cycles. (5.79) https://doi.org/10.1029/2018GB005880

Response: We thank the reviewer for the careful evaluation and helpful comments to improve the quality of the manuscript. We revised the manuscript and addressed the reviewer's comments, e.g., we make clearer the idea that atmospheric acidic processes associated with anthropogenic pollutants may transform unreactive P to bioavailable P, we cite Herbert et al., (2018) to support the idea, and etc.. In the special comments, there are three questions concerning these aspects. Please see the detailed responses to these comments below.

Specific comments: Line 16 of Abstract and elsewhere: The convention for what is

called in this manuscript DP, is actually TDP (Total Dissolved P). That is the P measured after persulphate oxidation in solution. Please change to TDP throughout.

Response: Revised as suggested.

Line 24: The authors suggest that humidity plays an important role in converting refractory P to bioavailable P. The most likely mechanism is that suggested in Nenes et al., (2010) which is the acidification of particles as they cycle from clouds, where the pH is rather high, to wet aerosols (where the pH is very low) and back again (see Stockdale et al., (2016).

Response: We added "This was likely caused by the acidification of particles as they cycled from cloud droplets to wet aerosols and back (Nenes et al., 2010; Stockdale et al., 2016)." in the Section 3.3.3 (Page 11, Lines 19-20).

Introduction Line 5 Add in offshore areas and regions where P limits. . ... General: Even in systems where N is the immediate limiting nutrient P can increase phytoplankton growth by moving the entire system to higher productvitiy.

Response: Revised as suggested.

Introduction page 2 line 9: The authors should comment/introduce the idea that anthropogenic processes can include the production of atmospheric acids, which can cause previously unreactive p to become bioavailable DIP. They discuss this possibility at length towards the end of their manuscript.

Response: In the revision, we added "In addition, atmospheric acidic processes associated with anthropogenic pollutants may transform unreactive P to bioavailable P. Recent model studies predict that acid dissolution process increases the fraction of bioavailable P from ~10% globally at labile pools to 42% in the Pacific Ocean, with the mean value of 22% in global marine atmosphere (Herbert et al., 2018).", and we cite Herbert et al., (2018) in the Introduction (Page 3, Lines 8-11).

Methods page 4 line 7 Remove 'in number of particles' and Replace monitored with

measured.

Response: Revised as suggested.

Page 5 line 19: What was the assumed value of Al in mineral dust that allowed the authors to assume that the particles were 8% by mass? I may have misunderstood what was written, in which case the authors should make it clearer.

Response: The description was revised to "The particle mass loading was estimated from Al contents by assuming that all aerosol Al was derived from mineral dust, which comprises 8 % mineral aerosol mass (Taylor, 1964)." (Page 5, Lines 23-25).

Page 5 line 25 What is 'floating' dust? A dust storm?

Response: Floating dust is a kind of dusty weather but is not in the stage of dust storms. It usually occurs after the passage of dust storms when there are still a considerable dust particles (floating) in the air. It is not a dust storm. We replaced "floating dust" with "dusty weather" in the revision.

Page 6 line 11 (and various other places including table 1) Aqaba is spelt wrongly. It is a b and not a d

Response: The typo was corrected.

Page 6 lines 31-34: If TP had high correlations with major elements (dust) and with heavy metals (anthropogenic) at the same time, is that not ambiguous?

Response: There are overlaps of mineral elements and heavy metals between natural dust (from desert) and anthropogenic particles (mainly from coal burning)

Page 7 line 3: The actual correlation data is not given (or at least not given here). This reviewer is a little confused as to what the authors mean by 'higher correlations' and whether that also means lower p values.

Response: The values were given in the supplementary materials. Please see Table

S1 in the Supplement.

Yes, "higher correlation" means having a larger correlation coefficient (r value) and a smaller p value. To avoid confusion, we modified the descriptions. Please see Lines 1-6 on Page 7.

Page 7 line 13 Are the authors convinced that soil dust (from deserts?) are an important source of DOP?

Response: Yes. Our results indicate that soil dust is one of substantial sources of DOP, in comparison with that in anthropogenic particles, and this is also partly the reason for the correlation between TDP and mineral elements. In addition, similar result was also reported by Myriokefalitakis et al. (2016), who estimated that the contribution of soil dust source to DOP was approximately 25 % on the global scale.

Page 7 Line 25 The authors might consider quoting Carbo et al., (2005) which presents the P solubility data for the Eastern Mediterranean in a more comprehensive manner than Herut et al. (2002). Carbo, P., Krom, M.D., Homoky, W.B., Benning, L.G., Herut, B., 2005. Impact of atmospheric deposition on N and P geochemistry in the southeastern Levantine basin. Deep-Sea Research II Volume 52: Nos 22-23, 3041-3053.

Response: We add Carbo et al (2005) in the revision.

Page 8 line 24: The data in that graph is non-linear

Response: Yes, they are non-linear, likely because of multiple reasons or mechanisms. If the data number is adequate to categorize them into different groups for a statistically meaning investigation, more detailed mechanisms could be addressed. Here, we can only show the trend, and the data show a statistically rough linear correlation (y=0.013x+75.8, R2=0.54, p<0.01).

Page 8 line 27 And because anthropogenic P is more likely to have interacted with pollutant gases to produce more bioavailable P

Response: We added this information "The reason is that anthropogenic P tends to associate loosely with particulates, dissolve more readily than mineral P, and, consequently, interact easily with acid gases to produce more bioavailable P (Herut et al., 2002; Baker et al., 2006a; 2006b; Anderson et al., 2010; Hsu et al., 2014; Herbert et al., 2018)." in the revised version. Please see Page 8, Lines 27-30.

Page 9 line 4 Remove obviously

Response: Removed in the revision.

Page 9 line 14 I had the same problem with Sholkovitz's paper too. It ignores the possibility that anthropogenic acids can interact with mineral dust to produce bioavailable P (or Fe). The authors of this article suggest this might be an important process themselves in line 31 "which more efficiently serves as a sink . . ..derived elements." And later on page 10 "Unfortunately we were unable to quantitatively distinguish the contributions of aerosol source and acidification to phosphorus solubility at this stage". That means both should be retained as possible sources. In reality the answer is probably that both more soluble P in anthropogenic particles at source and more P made soluble by acid processes in air masses from polluted sources occur and are in different proportions in different air masses.

Response: We totally agree with the reviewer. We discussed the contribution of aerosol sources to and the effects of atmospheric acidification process on P solubility, and also pointed out that both of them were two important factors/processes influencing the P solubility in the mentioned paragraph and pervious descriptions in the manuscript. We considered additional descriptions but feel such an addition would make the manuscript tedious. In order to avoid misunderstanding, we removed the sentence "Unfortunately we were unable to quantitatively distinguish the contributions of aerosol source and acidification to phosphorus solubility at this stage." in the revision.

Page 11 line 29 There seems to be a mistake in the first half of the line. I read it several times and could not decide what was meant.

Response: We revised the sentence as "On average, the P solubility was approximately 13 % in the aerosols of Panth/TP < 50 %, while the value was approximately 21 % in the aerosols of Panth/TP > 50 %." (Page 11, Lines 26-28).

Page 11 line 34 How did the authors define 'acidification degree of 150 nmol nmol-1? nmoles of what?

Response: We define the acidification degree with the molar ratio as [2nss-SO42-+NO3-]/TP. We add the information in the revision (Page 10, Line 22).

Page 12 line 3 Remove obviously

Response: Removed in the revision.

Page 12 Conclusions Very well written and this reviewer entirely agrees with the conclusions.

Thank you very much for your careful reading and helpful comments.

Please also note the supplement to this comment:
https://www.atmos-chem-phys-discuss.net/acp-2018-892/acp-2018-892-AC1-supplement.pdf

―――――――――――――――――

[Figure]

**Supplement:**

**Responses to Reviewer 1[#] Comments**

Reviewer #1 (Comments to Author):

General Comments:

I applaud the aim of this manuscript and I feel once it is modified slightly, that it will make an important contribution to the literature. I should say that a couple of years ago, we tried to do exactly the same data treatment using a data set collected in Crete. We had in total ~100 data points and we were unable to find significant patterns. This manuscript has 170 data points and has managed (just) to see some real patterns albeit the correlations they find are often statistically significant but with correlation coefficients of ~0.3!. In other general words, while the conclusions are interesting, they are not actually very strong.

In particular the authors seem to divide the particles into anthropogenic or dust only. They do not include the importance of acid processing of inorganic particles (dust or anthropogenic) as wet aerosols associated with clouds (high relative humidity) as a potentially important process. In fact they do discuss this in the text and state on page 10 that "Unfortunately , we were unable to quantitively distinguish the contributions of aerosol source and acidification to phosphorus solubility at this stage." Yet the text elsewhere minimises the possible contribution of acidification and emphasises instead anthropogenic particles which had high P solubility at source. This reviewer feels the manuscript would benefit by taking a more even balance between these two possibilities.

As a final general point, we have just published a paper in Global Biogeochemical Cycles (Herbert et al., 2018), which the authors obviously could not have seen. However it does predict that acid processes in China could be an important source of. Bioavailable P as a plume which passes over location such as Qingdoa and on to the western Pacific.

Herbert, Herbert R. J., Krom, M.D., Carslaw, K.S., Stockdale, A., Mortimer, R.J.G., Benning, L.G., Pringle, K., Browse, J., (2018) Quantifying the effect of atmospheric acid processing on the global deposition of bioavailable phosphorus from dust. Global Biogeochemical Cycles. (5.79) **https://doi.org/10.1029/2018GB005880**

Response: We thank the reviewer for the careful evaluation and helpful comments to improve the quality of the manuscript. We revised the manuscript and addressed the reviewer's comments, e.g., we make clearer the idea that atmospheric acidic processes associated with anthropogenic pollutants may transform unreactive P to bioavailable P, we cite Herbert et al., (2018) to support the idea, and etc.. In the special comments, there are three questions concerning these aspects. Please see the detailed responses to these comments below. The manuscript with the corresponding corrections marked yellow is attached to the end of this response letter.

Specific comments:
Line 16 of Abstract and elsewhere:
The convention for what is called in this manuscript DP, is actually TDP (Total Dissolved P). That is the P measured after persulphate oxidation in solution. Please change to TDP throughout.

Response: Revised as suggested.

Line 24: The authors suggest that humidity plays an important role in converting refractory P to bioavailable P. The most likely mechanism is that suggested in Nenes et al., (2010) which is the acidification of particles as they cycle from clouds, where the pH is rather high, to wet aerosols (where the pH is very low) and back again (see Stockdale et al., (2016).

Response: We added "This was likely caused by the acidification of particles as they cycled from cloud droplets to wet aerosols and back (Nenes et al., 2010; Stockdale et al., 2016)." in the Section 3.3.3 (Page 11, Lines 19-20).

Introduction
Line 5
Add in offshore areas and regions where P limits…..
General: Even in systems where N is the immediate limiting nutrient P can increase phytoplankton growth by moving the entire system to higher productvitiy.

Response: Revised as suggested.

Introduction page 2 line 9:
The authors should comment/introduce the idea that anthropogenic processes can include the production of atmospheric acids, which can cause previously unreactive p to become bioavailable DIP. They discuss this possibility at length towards the end of their manuscript.

Response: In the revision, we added "In addition, atmospheric acidic processes associated with anthropogenic pollutants may transform unreactive P to bioavailable P. Recent model studies predict that acid dissolution process increases the fraction of bioavailable P from ~10% globally at labile pools to 42% in the Pacific Ocean, with the mean value of 22% in global marine atmosphere (Herbert et al., 2018).", and we cite Herbert et al., (2018) in the Introduction (Page 3, Lines 8-11).

Methods page 4 line 7
Remove 'in number of particles' and Replace monitored with measured.

Response: Revised as suggested.

Page 5 line 19:
What was the assumed value of Al in mineral dust that allowed the authors to assume that the particles were 8% by mass? I may have misunderstood what was written, in which case the authors should make it clearer.

Response: The description was revised to "The particle mass loading was estimated from Al contents by assuming that all aerosol Al was derived from mineral dust, which comprises 8 % mineral aerosol mass (Taylor, 1964)." (Page 5, Lines 23-25).

Page 5 line 25

What is 'floating' dust? A dust storm?

Response: Floating dust is a kind of dusty weather but is not in the stage of dust storms. It usually occurs after the passage of dust storms when there are still a considerable dust particles (floating) in the air. It is not a dust storm. We replaced "floating dust" with "dusty weather" in the revision.

Page 6 line 11 (and various other places including table 1)
Aqaba is spelt wrongly. It is a b and not a d

Response: The typo was corrected.

Page 6 lines 31-34:
If TP had high correlations with major elements (dust) and with heavy metals (anthropogenic) at the same time, is that not ambiguous?

Response: There are overlaps of mineral elements and heavy metals between natural dust (from desert) and anthropogenic particles (mainly from coal burning)

Page 7 line 3:
The actual correlation data is not given (or at least not given here). This reviewer is a little confused as to what the authors mean by 'higher correlations' and whether that also means lower p values.

Response: The values were given in the supplementary materials. Please see Table S1 in the Supplement.

Yes, "higher correlation" means having a larger correlation coefficient ($r$ value) and a smaller $p$ value. To avoid confusion, we modified the descriptions. Please see Lines 1-6 on Page 7.

Page 7 line 13
Are the authors convinced that soil dust (from deserts?) are an important source of DOP?

Response: Yes. Our results indicate that soil dust is one of substantial sources of DOP, in comparison with that in anthropogenic particles, and this is also partly the reason for the correlation between TDP and mineral elements. In addition, similar result was also reported by Myriokefalitakis et al. (2016), who estimated that the contribution of soil dust source to DOP was approximately 25 % on the global scale.

Page 7 Line 25
The authors might consider quoting Carbo et al., (2005) which presents the P solubility data for the Eastern Mediterranean in a more comprehensive manner than Herut et al. (2002).
Carbo, P., Krom, M.D., Homoky, W.B., Benning, L.G., Herut, B., 2005. Impact of atmospheric deposition on N and P geochemistry in the southeastern Levantine basin. Deep-Sea Research II Volume 52: Nos 22-23, 3041-3053.

Response: We add Carbo et al (2005) in the revision.

Page 8 line 24:
The data in that graph is non-linear

Response: Yes, they are non-linear, likely because of multiple reasons or mechanisms. If the data number is adequate to categorize them into different groups for a statistically meaning investigation, more detailed mechanisms could be addressed. Here, we can only show the trend, and the data show a statistically rough linear correlation ($y=0.013x+75.8$, $R^2=0.54$, $p<0.01$).

Page 8 line 27
And because anthropogenic P is more likely to have interacted with pollutant gases to produce more bioavailable P

Response: We added this information "The reason is that anthropogenic P tends to associate loosely with particulates, dissolve more readily than mineral P, and, consequently, interact easily with acid gases to produce more bioavailable P (Herut et al., 2002; Baker et al., 2006a; 2006b; Anderson et al., 2010; Hsu et al., 2014; Herbert et al., 2018)." in the revised version. Please see Page 8, Lines 27-30.

Page 9 line 4
Remove obviously

Response: Removed in the revision.

Page 9 line 14
I had the same problem with Sholkovitz's paper too. It ignores the possibility that anthropogenic acids can interact with mineral dust to produce bioavailable P (or Fe). The authors of this article suggest this might be an important process themselves in line 31 "which more efficiently serves as a sink ….derived elements." And later on page 10 "Unfortunately we were unable to quantitatively distinguish the contributions of aerosol source and acidification to phosphorus solubility at this stage".
That means both should be retained as possible sources. In reality the answer is probably that both more soluble P in anthropogenic particles at source and more P made soluble by acid processes in air masses from polluted sources occur and are in different proportions in different air masses.

Response: We totally agree with the reviewer. We discussed the contribution of aerosol sources to and the effects of atmospheric acidification process on P solubility, and also pointed out that both of them were two important factors/processes influencing the P solubility in the mentioned paragraph and pervious descriptions in the manuscript. We considered additional descriptions but feel such an addition would make the manuscript tedious. In order to avoid misunderstanding, we removed the sentence "Unfortunately we were unable to quantitatively distinguish the contributions of aerosol source and acidification to phosphorus solubility at this stage." in the revision.

Page 11 line 29
There seems to be a mistake in the first half of the line. I read it several times and could not decide

what was meant.

Response: We revised the sentence as "On average, the P solubility was approximately 13 % in the aerosols of $P_{anth}$/TP < 50 %, while the value was approximately 21 % in the aerosols of $P_{anth}$/TP > 50 %." (Page 11, Lines 26-28).

Page 11 line 34 How did the authors define 'acidification degree of 150 nmol nmol-1? nmoles of what?

Response: We define the acidification degree with the molar ratio as $[2nss\text{-}SO_4^{2-}+NO_3^-]$/TP. We add the information in the revision (Page 10, Line 22).

Page 12 line 3
Remove obviously

Response: Removed in the revision.

Page 12 Conclusions
Very well written and this reviewer entirely agrees with the conclusions.

Thank you very much for your careful reading and helpful comments.

[revised manuscript text omitted]

---

## Author Comment (AC2) · 13 Dec 2018

Responses to Reviewer 2# Comments

Reviewer #2 (Comments to Author):

The manuscript investigated the P speciation and solubility in aerosols in the eastern China's coast. Phosphorus may ultimately control the primary production in the large areas of the ocean especially in the N-affluent regions such as the marginal seas of the western North Pacific. The previously reported P solubility in aerosols was in a wide range, and therefore it is important to understand the factors or mechanisms

determining the atmospheric input of soluble P. The manuscript studied the coordinated effect of relative humidity (RH) and aerosol origins and acidity on P solubility, and also included dissolved organic P in the discussion. The manuscript indicated that P in aerosols from anthropogenic sources had a higher solubility than the P in aerosols from mineral dust. Phosphorus solubility was usually less than 30% when the RH was below 60% and the higher RH increased the dissolution of aerosol P to a great degree under acidic conditions (how to define acidic condition?). These results will be very helpful to modeling the input of bioavailable P to the ocean. It would be nice if authors could discuss extrapolation of the results to the eastern China seas, other coastal regions or even the open ocean.

Response: We thank the reviewer for the careful evaluation and helpful comments to improve the quality of our manuscript. We revised the manuscript and addressed the reviewer's comments.

The acidic condition here is a relative value. We define it with respect to the average acidification degree of all samples.

We found that the P solubility and the Al concentration (as dust loading) displayed an inverse power-law relation (Section 3.3.1). Literature data from the eastern China sea areas also show similar relations between P solubility and mineral contents in aerosols. In the revision, we added "These results indicate the potential to extrapolate the discussion of this study to the seas to eastern China." on Page 9, Line 10.

Introduction Page 2 Line 3: Authors may add few sentences on the importance of atmospheric P deposition to the surface ocean. For example, long-term measurements of dissolved P at station ALOHA revealed unexpected temporal variability in PO43- concentrations in the surface ocean, which may be partly due to the episodic atmospheric deposition (Karl and Tien 1997). Karl DM, Tien G. 1997. Temporal variability in dissolved phosphorus concentrations in the subtropical North Pacific Ocean. Mar. Chem. 56:77–96

Response: In the revised version, we added "Atmospheric P deposition likely has an important contribution to phosphate and induces the growth of phytoplankton in surface seawater outside estuary areas, especially in offshore areas and regions where P limits phytoplankton growth (Karl and Tien, 1997; Paytan and McLaughlin, 2007; Mackey et al., 2012a)." in the Introduction (Page 2, Lines 4-6).

Methods Page 4 Line 26, "P" at the beginning of the sentence should be changed to "Phosphorus"

Response: Revised as suggested.

Page 5 Line 18: "because P is a substance in primary particles". Here what is the general size range for primary particles? Authors may provide the reference or the size distributions of P and DP to prove the statement. Page 5 line 18-20: "In cases when the samples contained less mineral dust, the aerosol mass would be somewhat under-estimated." The mass loadings estimated from Al concentrations may be compared to the officially reported PM10 concentrations to check for the average underestimation.

Response: The description of "because P is a substance in primary particles" in the original manuscript was changed to "because more than 90% of TP is in the >0.32 $\mu$m particles (Vicars et al., 2010)". Vicars et al (2010) was cited to support our statement. (Page 5, Line 23)

Vicars, W. C., Sickman, J. O., and Ziemann, P. J.: Atmospheric phosphorus deposition at a montane site: Size distribution, effects of wildfire, and ecological implications, Atmos. Environ., 44, 2813–2821, 2010.

The official PM10 data were available only after December 2013. We have a small number of filter-based TSP data, and they show a quite good correlation with the Al-based estimation. To avoid confusion, we deleted this sentence in the revision.

Results and Discussion Page 6 Line 3-5: The two sentences can be combined to be more concise.

Response: The two sentences have been combined to "DIP and DOP accounted, on average, for approximately 60 % and 40% of the TDP, respectively, indicating an appreciable contribution of DOP to the TDP." Please see Page 6, Lines 7-8.

Page 6 Line 27: "This result was probably caused by the release of primary biological particles and agriculture fertilization in spring." Was DOP released by the agricultural process in spring or as the loss of fertilizer (I thought that fertilizer should be mainly DIP)?

Response: Agricultural insecticides is one of sources of OP in atmospheric aerosols (Kanakidou et al., 2012). The description "agricultural fertilization" is inaccurate in the original manuscript, we replaced it with "agricultural activities" (Page 6, Line 30).

Page 8 Line 18: The statement "a small fraction of biological P" needs to be supported by a reference or observatory data.

Response: Wang et al. (2015) estimated combustion-related emissions of 1.8 Tg P yr-1, which represent over 50% of global atmospheric sources of P (3.5 Tg P yr-1), and the primary biogenic particles emission is 0.58 Tg P yr-1 to contribute 17% of the total P emission. We cited this reference to support our statement (Page 8, Line 23).

Wang, R., Balkanski, Y., Boucher, O., Ciais, P., Penuelas, J., and Tao, S.: Significant contribution of combustion-related emissions to the atmospheric phosphorus budget, Nat. Geosci., 8, 48–54, 2015.

Page 8 Line 29-31: Why is the correlation of TDP vs soluble Al better than that of TP vs Al? Such comparison is hard to explain. Authors may delete this sentence and just compare the ratios of P/Al to the ratios of TDP/soluble Al.

Response: We deleted this sentence as suggested.

Page 9 Line 31: "Some data points deviated from the fitting curves." The specific variables for fitting curves should be indicated here, e.g. for P solubility and total Al.

Response: We added the information of the specific variables in the revision. This sentence was revised as "Some data points, for P solubility against total Al and the specific surface area, deviated from the fitting curves (Fig. 4, Fig. 5), . . . . . ." (Page 10, Lines 3-4)

Page 10 Line 11-13 & 18-20: There is repetition in these sentences.

Response: The sentence repeated on Page 10 Line 11-13 in the previous manuscript have been revised to "Following Hsu et al. (2014) who used the ratio of acids/total Fe to investigate the influence of aerosol acidification on the Fe solubility, we use the ratio of acids/total P to investigate the influence of aerosol acidification on the P solubility." Please see Page 10, Lines 16-17.

The two sentences with repetition on Page 10 Line 18-20 in the original manuscript were revised to "However, the P solubility versus the acid/TP ratio followed different regression curves corresponding to the ranges of Panth/TP, e.g., the slope of 0.22 in samples with Panth/TP > 70 %, while of 0.01 in samples with Panth/TP < 50 %." Please see Page 10, Lines 22-24.

Page 11 Line 6: "had a statistically significant correlation" What are the variables?

Response: We added the information of the variables. This sentence was revised as "The data points of P solubility against acids/total P for the samples of 50 % < Panth/TP < 70 % were frequently between the two fitted curves of Panth/TP > 70 % and Panth/TP < 50 % (Fig. 6), and had a statistically significant correlation at 99 % confidence (r = 0.383, p = 0.006)." (Page 11, Lines 11-13)

Page 11 Line 17-32: The coordinate effect could be arranged as another section. The relationships between P solubility and humidity, anthropogenic percentage and acidification are complex. The two paragraphs seem to talk about the situations at RH<60% and RH>60% respectively. But the RH change from <60% to >60% was discussed again in the second paragraph. This part needs to be reorganized.

Response: The first paragraph of Section 3.3.3 demonstrates the effect of relative humidity on P solubility, and the latter two paragraphs demonstrate the coordinate effect of relative humidity and atmospheric acidification. In the revised version, we changed the title of this section to "Relative humidity and coordinate effect of aerosol sources and acidity ". (Page 11, Line 10)

We discuss P solubility corresponding to aerosol sources and acidity at relative humidity < 60% and > 60%, respectively. In the part associated with relative humidity > 60%, we compared the changes of P solubility with the relative humidity corresponding to aerosol sources and acidities. The discussion of relative humidity from <60% to >60% is remained in the manuscript because we effort to compare the increase of the P solubility when the relative humidity was > 60% with that when the relative humidity was < 60%.

Page 11 Line 25: The unit of acidification degree should be unified in the paper. Is it proper to choose 150 acidification degree as the boundary?

Response: The unit of acidification degree was unified in the revised version.

The acidification degree of 150 nmol nmol-1 was close to the average acidification degree of all aerosol samples. We chose this value as the boundary to demonstrate the difference between the two groups of samples above and below the average acidification degree.

Page 12 Line 1-2: The first half sentence talked about the effect of RH on P solubility, and the second half mentioned acidification. The linkage between the RH and acidification was missing.

Response: This sentence has been revised as "Overall, the enhancement of P solubility by RH increase in the mineral aerosols was much lower than in the anthropogenic aerosols, in which P was more susceptible to acidification, e.g., at RH>60%.". Please see Page 12 Lines 6-7.

Conclusion Page 12 Line 12: ".….from mineral dust and anthropogenic sources" can be deleted.

Response: Revised as suggested.

Page 12 Line 17: "The threshold RH for this effect was approximately 60%." can be deleted.

Response: Revised as suggested.

Please also note the supplement to this comment:
https://www.atmos-chem-phys-discuss.net/acp-2018-892/acp-2018-892-AC2-supplement.pdf

**Supplement:**

**Responses to Reviewer 2# Comments**

Reviewer #2 (Comments to Author):

The manuscript investigated the P speciation and solubility in aerosols in the eastern China's coast. Phosphorus may ultimately control the primary production in the large areas of the ocean especially in the N-affluent regions such as the marginal seas of the western North Pacific. The previously reported P solubility in aerosols was in a wide range, and therefore it is important to understand the factors or mechanisms determining the atmospheric input of soluble P. The manuscript studied the coordinated effect of relative humidity (RH) and aerosol origins and acidity on P solubility, and also included dissolved organic P in the discussion. The manuscript indicated that P in aerosols from anthropogenic sources had a higher solubility than the P in aerosols from mineral dust. Phosphorus solubility was usually less than 30% when the RH was below 60% and the higher RH increased the dissolution of aerosol P to a great degree under acidic conditions (how to define acidic condition?). These results will be very helpful to modeling the input of bioavailable P to the ocean. It would be nice if authors could discuss extrapolation of the results to the eastern China seas, other coastal regions or even the open ocean.

Response: We thank the reviewer for the careful evaluation and helpful comments to improve the quality of our manuscript. We revised the manuscript and addressed the reviewer's comments. The manuscript with the corresponding corrections marked yellow is attached to the end of this response letter.

The acidic condition here is a relative value. We define it with respect to the average acidification degree of all samples.

We found that the P solubility and the Al concentration (as dust loading) displayed an inverse power-law relation (Section 3.3.1). Literature data from the eastern China sea areas also show similar relations between P solubility and mineral contents in aerosols. In the revision, we added "These results indicate the potential to extrapolate the discussion of this study to the seas to eastern China." on Page 9, Line 10.

Introduction Page 2 Line 3: Authors may add few sentences on the importance of atmospheric P deposition to the surface ocean. For example, long-term measurements of dissolved P at station ALOHA revealed unexpected temporal variability in $PO_4^{3-}$ concentrations in the surface ocean, which may be partly due to the episodic atmospheric deposition (Karl and Tien 1997).
Karl DM, Tien G. 1997. Temporal variability in dissolved phosphorus concentrations in the subtropical North Pacific Ocean. Mar. Chem. 56:77–96

Response: In the revised version, we added "Atmospheric P deposition likely has an important contribution to phosphate and induces the growth of phytoplankton in surface seawater outside estuary areas, especially in offshore areas and regions where P limits phytoplankton growth (Karl and Tien, 1997; Paytan and McLaughlin, 2007; Mackey et al., 2012a)." in the Introduction (Page 2, Lines 4-6).

Methods Page 4 Line 26, "P" at the beginning of the sentence should be changed to "Phosphorus"

Response: Revised as suggested.

Page 5 Line 18: "because P is a substance in primary particles". Here what is the general size range for primary particles? Authors may provide the reference or the size distributions of P and DP to prove the statement. Page 5 line 18-20: "In cases when the samples contained less mineral dust, the aerosol mass would be somewhat underestimated." The mass loadings estimated from Al concentrations may be compared to the officially reported PM10 concentrations to check for the average underestimation.

Response: The description of "because P is a substance in primary particles" in the original manuscript was changed to "because more than 90% of TP is in the >0.32 μm particles (Vicars et al., 2010)". Vicars et al (2010) was cited to support our statement. (Page 5, Line 23)

Vicars, W. C., Sickman, J. O., and Ziemann, P. J.: Atmospheric phosphorus deposition at a montane site: Size distribution, effects of wildfire, and ecological implications, Atmos. Environ., 44, 2813–2821, 2010.

The official $PM_{10}$ data were available only after December 2013. We have a small number of filter-based TSP data, and they show a quite good correlation with the Al-based estimation. To avoid confusion, we deleted this sentence in the revision.

Results and Discussion Page 6 Line 3-5: The two sentences can be combined to be more concise.

Response: The two sentences have been combined to "DIP and DOP accounted, on average, for approximately 60 % and 40% of the TDP, respectively, indicating an appreciable contribution of DOP to the TDP." Please see Page 6, Lines 7-8.

Page 6 Line 27: "This result was probably caused by the release of primary biological particles and agriculture fertilization in spring." Was DOP released by the agricultural process in spring or as the loss of fertilizer (I thought that fertilizer should be mainly DIP)?

Response: Agricultural insecticides is one of sources of OP in atmospheric aerosols (Kanakidou et al., 2012). The description "agricultural fertilization" is inaccurate in the original manuscript, we replaced it with "agricultural activities" (Page 6, Line 30).

Page 8 Line 18: The statement "a small fraction of biological P" needs to be supported by a reference or observatory data.

Response: Wang et al. (2015) estimated combustion-related emissions of 1.8 Tg P $yr^{-1}$, which represent over 50% of global atmospheric sources of P (3.5 Tg P $yr^{-1}$), and the primary biogenic particles emission is 0.58 Tg P $yr^{-1}$ to contribute 17% of the total P emission. We cited this reference to support our statement (Page 8, Line 23).

Wang, R., Balkanski, Y., Boucher, O., Ciais, P., Penuelas, J., and Tao, S.: Significant contribution of combustion-related emissions to the atmospheric phosphorus budget, Nat. Geosci., 8, 48–54, 2015.

Page 8 Line 29-31: Why is the correlation of TDP vs soluble Al better than that of TP vs Al? Such comparison is hard to explain. Authors may delete this sentence and just compare the ratios of P/Al to

the ratios of TDP/soluble Al.

Response: We deleted this sentence as suggested.

Page 9 Line 31: "Some data points deviated from the fitting curves." The specific variables for fitting curves should be indicated here, e.g. for P solubility and total Al.

Response: We added the information of the specific variables in the revision. This sentence was revised as "Some data points, for P solubility against total Al and the specific surface area, deviated from the fitting curves (Fig. 4, Fig. 5), ……" (Page 10, Lines 3-4)

Page 10 Line 11-13 & 18-20: There is repetition in these sentences.

Response: The sentence repeated on Page 10 Line 11-13 in the previous manuscript have been revised to "Following Hsu et al. (2014) who used the ratio of acids/total Fe to investigate the influence of aerosol acidification on the Fe solubility, we use the ratio of acids/total P to investigate the influence of aerosol acidification on the P solubility." Please see Page 10, Lines 16-17.

The two sentences with repetition on Page 10 Line 18-20 in the original manuscript were revised to "However, the P solubility versus the acid/TP ratio followed different regression curves corresponding to the ranges of $P_{anth}$/TP, e.g., the slope of 0.22 in samples with $P_{anth}$/TP > 70 %, while of 0.01 in samples with $P_{anth}$/TP < 50 %." Please see Page 10, Lines 22-24.

Page 11 Line 6: "had a statistically significant correlation" What are the variables?

Response: We added the information of the variables. This sentence was revised as "The data points of P solubility against acids/total P for the samples of 50 % < $P_{anth}$/TP < 70 % were frequently between the two fitted curves of $P_{anth}$/TP > 70 % and $P_{anth}$/TP < 50 % (Fig. 6), and had a statistically significant correlation at 99 % confidence ($r = 0.383$, $p = 0.006$)." (Page 11, Lines 11-13)

Page 11 Line 17-32: The coordinate effect could be arranged as another section. The relationships between P solubility and humidity, anthropogenic percentage and acidification are complex. The two paragraphs seem to talk about the situations at RH<60% and RH>60% respectively. But the RH change from <60% to >60% was discussed again in the second paragraph. This part needs to be reorganized.

Response: The first paragraph of Section 3.3.3 demonstrates the effect of relative humidity on P solubility, and the latter two paragraphs demonstrate the coordinate effect of relative humidity and atmospheric acidification. In the revised version, we changed the title of this section to "Relative humidity and coordinate effect of aerosol sources and acidity ". (Page 11, Line 10)

We discuss P solubility corresponding to aerosol sources and acidity at relative humidity < 60% and > 60%, respectively. In the part associated with relative humidity > 60%, we compared the changes of P solubility with the relative humidity corresponding to aerosol sources and acidities. The discussion of relative humidity from <60% to >60% is remained in the manuscript because we effort to compare the

increase of the P solubility when the relative humidity was > 60% with that when the relative humidity was < 60%.

Page 11 Line 25: The unit of acidification degree should be unified in the paper. Is it proper to choose 150 acidification degree as the boundary?

Response: The unit of acidification degree was unified in the revised version.

The acidification degree of 150 nmol nmol$^{-1}$ was close to the average acidification degree of all aerosol samples. We chose this value as the boundary to demonstrate the difference between the two groups of samples above and below the average acidification degree.

Page 12 Line 1-2: The first half sentence talked about the effect of RH on P solubility, and the second half mentioned acidification. The linkage between the RH and acidification was missing.

Response: This sentence has been revised as "Overall, the enhancement of P solubility by RH increase in the mineral aerosols was much lower than in the anthropogenic aerosols, in which P was more susceptible to acidification, e.g., at RH>60%.". Please see Page 12 Lines 6-7.

Conclusion Page 12 Line 12: ".....from mineral dust and anthropogenic sources" can be deleted.

Response: Revised as suggested.

Page 12 Line 17: "The threshold RH for this effect was approximately 60%." can be deleted.

Response: Revised as suggested.

[revised manuscript text omitted]

---

## Author Response (AR2)

Dear Prof. Maria Kanakidou,

Thank you very much for your handling our submission and the evaluation and helpful comments to improve the quality of our manuscript.

We have revised the manuscript regarding the latest comments. Details of the point-by-point responses to the comments and the corresponding revisions in the manuscript are described in the responding letter. The manuscript with the corresponding corrections marked yellow is attached to the end of this cover letter.

Sincerely,

Jinhui Shi
College of Environmental Science and Engineering
Ocean University of China
Qingdao, 266100
P.R. China

**Responses to Reviewer's Comments**

**Co-Editor Decision: Publish subject to minor revisions (review by editor)** (27 Dec 2018) by Maria Kanakidou
Comments to the Author:
Thank you very much for the thorough revision of your manuscript. There are some additional corrections needed before the manuscript can be accepted for publication in ACP.

1- First of all the most appropriate naming for 'Panth' is 'Pnondust' since what is calculated by equation (2) is the residual P once the P from dust is removed based on Al concentrations in the aerosols. In addition, there are cases where aerosols of biological origin (page 11, line 4) or even one could claim sea-salt aerosol that contains P and Cl- (for instance how the points with high Cl-to-TP ratio in Figure 6c are explained?)

Response: $P_{anth}$ is revised to $P_{nd}$ ($P_{non-dust}$) as suggested.

In the revision, we added "Kanakidou et al (2012) provided an organic phosphorus budget showing a relatively large marine source (almost 80% with large uncertainties in magnitude). The organic P in the two samples with high Cl-/TP molar ratio might be, at least partly, caused by the contribution from marine biological sources besides terrestrial biological sources." (Page 11, Lines 7-10).

2- Concerning the DOP from soil dust (page 7, 2nd paragraph and abstract line 28):
What the authors show with their data is that high DOP values are associated with the presence of crustal elements in the aerosol; the correlation with crustal elements does not necessarily mean that soil

dust is a source of DOP. The coexistence of bioaerosols with dust under intensive dust events has been observed in the Mediterranean and in the Atlantic (see for instance Griffin et al Atmos. Environ. 2007 and discussion in Kanakidou et al. Environmental Research Letters, 2018).

Therefore I would argue that the observed correlation between DOP and crustal elements could indicate the co-existence of bioaerosols rich in organic phosphorus and of dust with crustal material.

Also Kanakidou et al (2012) provided an organic phosphorus budget showing a relatively large marine source (almost 80%; but very uncertain in magnitude) and for the continental sources contributions of 4.5% (soil dust), 9% (bioaerosols) and 7% (fossil fuel and biomass burning).

Furthermore, Myriokefalitakis et al (2016) reported that, at the global scale, approximately half of atmospheric TDP (and not DOP as stated in the manuscript page7 line 15) is from primary biological aerosol particles. Myriokefalitakis et al (2016) also provided budgets (in their supplementary material) that show contributions of various sources to the total source of TDP (referred as DP in their manuscript) : about 39% from soil dust, 3.5% from biomass burning, 8% from anthropogenic sources and 45% from bioaerosols. Note that these numbers are associated with large uncertainty.

Please correct the numbers and the discussion accordingly.

Griffin DW, Kubilay N, Kocak M, Gray M A, Borden T C and Shinn E A 2007 Airborne desert dust and aeromicrobiology over the Turkish Mediterranean coastline Atmos. Environ. 41, 4050–62

Kanakidou M., Myriokefalitakis S., Tsigaridis K.: Aerosols in atmospheric chemistry and biogeochemical cycles of nutrients, Environ. Res. Lett. 13 063004, 2018. https://doi.org/10.1088/1748-9326/aabcdb

Response: We thank the reviewer for the careful comments, and we totally agree with these comments. In the revision, we added the sentence "In addition, bioaerosols were enriched in long-range transported dust plumes in Beijing and the Northwestern Pacific for Asian dust and in the Mediterranean and the Atlantic for African dust (Yuan et al., 2017; Hara and Zhang, 2012; Griffin et al., 2007; Kanakidou et al., 2018). Therefore, the observed correlation between DOP and crustal elements could be the consequence of the presence of bioaerosols in dust plumes." in the revision. The references of Griffin et al. (2007), Kanakidou et al. (2018) for African dust, and also Yuan et al. (2017) and Hara and Zhang (2012) for Asian dust were added (Page 7, Lines 17-20).

Yuan, H., Zhang, D., Shi, Y., Li, B., Yang, J., Yu, X., Chen, N., and Kakikawa M.: Cell concentration, viability and culture composition of airborne bacteria during a dust event in Beijing, J. Environ. Sci., 55, 33-40, 2017.

Hara, K., and Zhang, D.: Bacterial abundance and viability in long-range transported dust, Atmos. Environ., 47, 20-25, 2012.

The reference of Kanakidou et al (2012) providing the organic phosphorus budget is cited in the revision (Page 11, Lines 7-8).

The sentence "Myriokefalitakis et al. (2016) reported that, at the global scale, approximately 50 % of

DOP is from primary biological aerosol particles, ……" was not accurately described in the original manuscript. We intended quoting the data in Table 3 of the reference (Myriokefalitakis et al., 2016), which is "Secondary DP sources (in Tg-P yr$^{-1}$) due to OP ageing contained in biomass burning, anthropogenic combustion, sea spray and dust as well as due to dust (apatite) dissolution via the acid-solubilization mechanism". In the revision, we modified this sentence into "Myriokefalitakis et al. (2016) reported that, on the global scale, approximately 50 % of secondary DOP was attributed to the ageing of biological P primarily contained in aerosols, and the contributions of soil dust, anthropogenic combustion and biomass burning ageing to secondary DOP were approximately 25 %, 15.6 % and 9.4 %, respectively." Please see Page 7, Lines 14-17.

3- Page 2, line 31: remove 'oceanic' this range refers to the global scale.

Response: Removed in the revision.

4- Page 2, line 32: add Myriokefalitakis et al., 2016 reference there also.

Response: Added the reference as suggested.

5- Page 3, line 11: Myriokefalitakis et al. (2016) estimated that acid-driven solubilisation flux of P from mineral dust contributed about one third to the total TDP source.

Response: In the revision, we added "Myriokefalitakis et al. (2016) estimated that acid-driven solubilization flux of P from mineral dust contributed about one third to the global TDP atmospheric source." (Page 3, lines 11-12).

6- Page 5, lines 24-25: ' which comprises 8% mineral aerosol mass' do you mean : ' in which Al comprises 8% of the mineral aerosol mass' ? please check phrasing.

Response: Yes. The description was revised to "……, in which Al comprises 8% of the mineral aerosol mass (Taylor, 1964)." (page 5, lines 24-25).

7- Page 7, line 26: add also reference by Nenes et al., 2011

Response: We added Nenes et al (2011) in the revision.

We express again our appreciation to the reviewers and the co-editor for their comments and suggestions, which have helped us to greatly increase the readership of this paper. In the *Acknowledgments*, "
[revised manuscript text omitted]